# Aligning Diffusion Behaviors with Q-functions for Efficient Continuous Control

**Huayu Chen**[1,2], **Kaiwen Zheng**[1,2], **Hang Su**[1,2,3], **Jun Zhu**[1,2,3*]
[1]Department of Computer Science and Technology, Tsinghua University
[2]Institute for AI, BNRist Center, Tsinghua-Bosch Joint ML Center, THBI Lab, Tsinghua University
[3]Pazhou Lab (Huangpu), Guangzhou, China

## Abstract

Drawing upon recent advances in language model alignment, we formulate offline Reinforcement Learning as a two-stage optimization problem: First pretraining expressive generative policies on reward-free behavior datasets, then fine-tuning these policies to align with task-specific annotations like Q-values. This strategy allows us to leverage abundant and diverse behavior data to enhance generalization and enable rapid adaptation to downstream tasks using minimal annotations. In particular, we introduce Efficient Diffusion Alignment (EDA) for solving continuous control problems. EDA utilizes diffusion models for behavior modeling. However, unlike previous approaches, we represent diffusion policies as the derivative of a scalar neural network with respect to action inputs. This representation is critical because it enables direct density calculation for diffusion models, making them compatible with existing LLM alignment theories. During policy fine-tuning, we extend preference-based alignment methods like Direct Preference Optimization (DPO) to align diffusion behaviors with continuous Q-functions. Our evaluation on the D4RL benchmark shows that EDA exceeds all baseline methods in overall performance. Notably, EDA maintains about 95% of performance and still outperforms several baselines given only 1% of Q-labelled data during fine-tuning. Code: `https://github.com/thu-ml/Efficient-Diffusion-Alignment`

## 1 Introduction

Learning diverse behaviors is generative modeling; transforming them into optimized policies is reinforcement learning. Recent studies have identified diffusion policies as a powerful tool for representing heterogeneous behavior datasets [21, 38]. However, these behavior policies incorporate suboptimal decisions in datasets, making them unsuitable for direct deployment in downstream tasks. To get optimized policies, typical methods involve either augmenting the behavior policy with an additional guidance/evaluation network [21, 32, 15] or training a new evaluation policy supervised by the behavior policy [13, 4]. While functional, these methods fail to leverage the full potential of pretrained behaviors as they require constructing some new policy models from scratch. This raises the question: *Can we directly fine-tune pretrained diffusion behaviors into optimized policies?*

Recent advances in Large Language Model (LLM) alignment techniques [57, 37, 43] offer valuable insights for fine-tuning diffusion behavior policies due to the fundamental similarity of the issues they aim to address (Fig. 1). While pretrained LLMs accurately imitate language patterns from web-scale corpus, they also capture toxic or unwanted content within the dataset. Alignment algorithms, such as Direct Preference Optimization (DPO, [41]), are designed to remove harmful or useless content learned during pretraining. They enable quick adaptations of pretrained LLMs to human intentions

---

*The corresponding author.

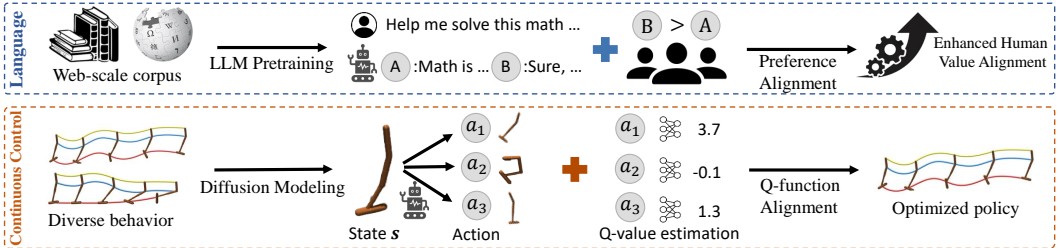

Figure 1: Comparison between alignment strategies for LLMs and diffusion policies (ours).

by fine-tuning them on a small dataset annotated with human preference labels. These strategies, due to their simplicity and effectiveness, have seen widespread applications in academia and industry.

Despite the high similarity in problem formulation and the immense potential of LLM alignment techniques, they cannot be directly applied to fine-tune diffusion policy in domains like continuous control. This is primarily because LLMs employ Categorical models to deal with discrete actions (tokens). Their alignment relies on computing data probabilities for maximum likelihood training (Sec. 2.2). However, diffusion models lack a tractable probability calculation method in continuous action space [5]. Additionally, the data annotation method differs significantly between two areas: LM alignment uses binary preference labels for comparing responses, while continuous control uses scalar Q-functions for evaluating actions (Fig. 1).

To allow aligning diffusion behavior models with Q-functions for policy optimization, we introduce Efficient Diffusion Alignment (EDA). EDA consists of two stages: behavior pretraining and policy fine-tuning. During the pretraining stage, we learn a conditional diffusion behavior model on reward-free datasets. Different from previous work which constructs diffusion models as an end-to-end network, we represent diffusion policies as the derivative of a scalar neural network with respect to action inputs. This representation is critical because it enables direct density calculation for diffusion policies. We demonstrate that the scalar network exactly outputs the unnormalized density of behavior distributions, making diffusion policies compatible with existing LLM alignment theories.

During the fine-tuning stage, we propose a novel algorithm that directly fine-tunes pretrained behavior models into optimized diffusion policies. The training objective is strictly derived by constructing a classification task to predict the optimal action using log-probability ratios between the policy and the behavior model. Our approach innovatively expands DPO by allowing fine-tuning on an arbitrary number of actions annotated with explicit Q-values, beyond just the typical binary preference data.

One main advantage of EDA is that it enables fast and data-efficient adaptations of behavior models in downstream tasks. Our experiments on the D4RL benchmark [10] show that EDA maintains 95 % of its performance and still surpasses baselines like IQL [25] with just 1% of Q-labelled data relative to the pretraining phase. Besides, EDA exhibits rapid convergence during fine-tuning, requiring only about 20K gradient steps (about 2% of the typical 1M policy training steps) to achieve convergence. Finally, EDA outperforms all reference baselines in overall performance with access to the full datasets. We attribute the high efficiency of EDA to its exploitation of the diffusion behavior models' generalization ability acquired during pretraining.

Our key contributions: 1. We represent diffusion policies as the derivative of a scalar value network to allow direct density estimation. This makes diffusion policies compatible with the existing alignment framework. 2. We extend preference-based alignment methods and propose EDA to align diffusion behaviors with scalar Q-functions, showcasing its vast potential in continuous control.

## 2 Background

### 2.1 Offline Reinforcement Learning

Offline RL aims to tackle decision-making problems by solely utilizing a pre-collected behavior dataset. Consider a typical Markov Decision Process (MDP) described by the tuple $\langle \mathcal{S}, \mathcal{A}, P, r, \gamma \rangle$. $\mathcal{S}$ is the state space, $\mathcal{A}$ is the action space, $P(\boldsymbol{s}'|\boldsymbol{s}, \boldsymbol{a})$ is the transition function, $r(\boldsymbol{s}, \boldsymbol{a})$ is the reward function and $\gamma$ is the discount factor. Given a static dataset $\mathcal{D}^{\mu} := \{\boldsymbol{s}, \boldsymbol{a}, r, \boldsymbol{s}'\}$ representing

interaction history between an implicit policy $\mu$ and the MDP environment, our goal is to learn a new policy that maximizes cumulative rewards in this MDP while staying close to the behavior policy $\mu$.

Offline RL can be formalized as a constrained policy optimization problem [26, 35, 53]:

$$\max_{\pi} \mathbb{E}_{\boldsymbol{s} \sim \mathcal{D}^{\mu}, \boldsymbol{a} \sim \pi(\cdot|\boldsymbol{s})} Q(\boldsymbol{s}, \boldsymbol{a}) - \beta D_{\mathrm{KL}} \left[ \pi(\cdot|\boldsymbol{s}) || \mu(\cdot|\boldsymbol{s}) \right], \tag{1}$$

where $Q(\boldsymbol{s}, \boldsymbol{a})$ is an action evaluation network that can be learned from $\mathcal{D}^{\mu}$. $\beta$ is a temperature coefficient. Previous work [40, 39] proves that the optimal solution for solving Eq. 1 is:

$$\pi^*(\boldsymbol{a}|\boldsymbol{s}) = \frac{1}{Z(\boldsymbol{s})} \mu(\boldsymbol{a}|\boldsymbol{s}) e^{Q(\boldsymbol{s}, \boldsymbol{a})/\beta}. \tag{2}$$

In this paper, we focus on how to efficiently learn parameterized policies for modeling $\pi^*$.

## 2.2 Direct Preference Optimization for Language Model Alignment

Direct Preference Optimization (DPO, [41]) is a fine-tuning technique for aligning pretrained LLMs with human feedback. Suppose we already have a pretrained LLM model $\mu(\boldsymbol{a}|\boldsymbol{s})$, where $\boldsymbol{s}$ represents user instructions, and $\boldsymbol{a}$ represents generated responses. The goal is to align $\mu_{\phi}$ with some implicit evaluation rewards $r^{\mathrm{LM}}(\boldsymbol{s}, \boldsymbol{a})$ that reflect human preference. Our target model is $\pi^*(\boldsymbol{a}|\boldsymbol{s}) \propto \mu_{\phi}(\boldsymbol{a}|\boldsymbol{s}) e^{r^{\mathrm{LM}}(\boldsymbol{s}, \boldsymbol{a})/\beta}$.

DPO assumes we only have access to some pairwise preference data $\{\boldsymbol{s} \to (\boldsymbol{a}_w > \boldsymbol{a}_l)\}$ and the preference probability is influenced by $r^{\mathrm{LM}}(\boldsymbol{s}, \boldsymbol{a})$. Formally,

$$p(\boldsymbol{a}_w \succ \boldsymbol{a}_l | \boldsymbol{s}) := \frac{e^{r^{\mathrm{LM}}(\boldsymbol{s}, \boldsymbol{a}_w)}}{e^{r^{\mathrm{LM}}(\boldsymbol{s}, \boldsymbol{a}_l)} + e^{r^{\mathrm{LM}}(\boldsymbol{s}, \boldsymbol{a}_w)}} = \sigma(r^{\mathrm{LM}}(\boldsymbol{s}, \boldsymbol{a}_w) - r^{\mathrm{LM}}(\boldsymbol{s}, \boldsymbol{a}_l)), \tag{3}$$

where $\sigma$ is the sigmoid function.

In order to learn $\pi_{\theta} \approx \pi^*(\boldsymbol{a}|\boldsymbol{s}) \propto \mu_{\phi}(\boldsymbol{a}|\boldsymbol{s}) e^{r^{\mathrm{LM}}(\boldsymbol{s}, \boldsymbol{a})/\beta}$, DPO first parameterizes a reward model using the log-probability ratio between $\pi_{\theta}$ and $\mu_{\phi}$, and then optimizes this reward model through maximum likelihood training:

$$\mathcal{L}_{\mathrm{DPO}} = -\mathbb{E}_{\{\boldsymbol{s}, \boldsymbol{a}_w \succ \boldsymbol{a}_l\}} \log \sigma(r_{\theta}^{\mathrm{LM}}(\boldsymbol{s}, \boldsymbol{a}_w) - r_{\theta}^{\mathrm{LM}}(\boldsymbol{s}, \boldsymbol{a}_l)), \tag{4}$$

$$\text{where} \qquad r_{\theta}^{\mathrm{LM}}(\boldsymbol{s}, \boldsymbol{a}) := \beta \log \frac{\pi_{\theta}(\boldsymbol{a}|\boldsymbol{s})}{\mu_{\phi}(\boldsymbol{a}|\boldsymbol{s})}$$

The key insight behind DPO's loss function is the equivalence and mutual convertibility between the policy model and the reward model. This offers a new perspective for solving generative policy optimization problems by applying discriminative classification loss.

## 2.3 Diffusion Modeling for Estimating Behavior Score Functions

Recent studies show that diffusion models [45, 20, 49] excel at representing heterogeneous behavior policies in continuous control [21, 5, 38]. To train a diffusion behavior model, we first gradually inject Gaussian noise into action points according to the forward diffusion process:

$$\boldsymbol{a}_t = \alpha_t \boldsymbol{a} + \sigma_t \boldsymbol{\epsilon}, \tag{5}$$

where $t \in [0, 1]$, and $\boldsymbol{\epsilon}$ is standard Gaussian noise. $\alpha_t, \sigma_t \in [0, 1]$ are manually defined so that at time $t = 0$, we have $\boldsymbol{a}_t = \boldsymbol{a}$ and at time $t = 1$, we have $\boldsymbol{a}_t \approx \boldsymbol{\epsilon}$. When $\boldsymbol{a}$ is sampled from the behavior policy $\mu(\boldsymbol{a}|\boldsymbol{s})$, the marginal distribution of $\boldsymbol{a}_t$ at various time $t$ satisfies

$$\mu_t(\boldsymbol{a}_t|\boldsymbol{s}, t) = \int \mathcal{N}(\boldsymbol{a}_t|\alpha_t \boldsymbol{a}, \sigma_t^2 \boldsymbol{I}) \mu(\boldsymbol{a}|\boldsymbol{s}, t) \mathrm{d}\boldsymbol{a}. \tag{6}$$

Intuitively, the diffusion training objective predicts the noise added to the original behavior actions:

$$\min_{\phi} \mathbb{E}_{t, \boldsymbol{\epsilon}, \boldsymbol{s}, \boldsymbol{a} \sim \mu(\cdot|\boldsymbol{s})} \left[ ||\boldsymbol{\epsilon}_{\phi}(\boldsymbol{a}_t|\boldsymbol{s}, t) - \boldsymbol{\epsilon}||_2^2 \right]_{\boldsymbol{a}_t = \alpha_t \boldsymbol{a} + \sigma_t \boldsymbol{\epsilon}}. \tag{7}$$

More formally, it can be proved that the learned "noise predictor" $\boldsymbol{\epsilon}_{\phi}$ actually represents the *score function* $\nabla_{\boldsymbol{a}_t} \log \mu_t(\boldsymbol{a}_t|\boldsymbol{s}, t)$ of the diffused behavior distribution $\mu_t$ [49]:

$$\nabla_{\boldsymbol{a}_t} \log \mu_t(\boldsymbol{a}_t|\boldsymbol{s}, t) = -\boldsymbol{\epsilon}^*(\boldsymbol{a}_t|\boldsymbol{s}, t)/\sigma_t \approx -\boldsymbol{\epsilon}_{\phi}(\boldsymbol{a}_t|\boldsymbol{s}, t)/\sigma_t. \tag{8}$$

With such a score-function estimator, we can employ existing numerical solvers [46, 33] to reverse the diffusion process, and sample actions from the learned behavior policy $\mu_{\phi}$.

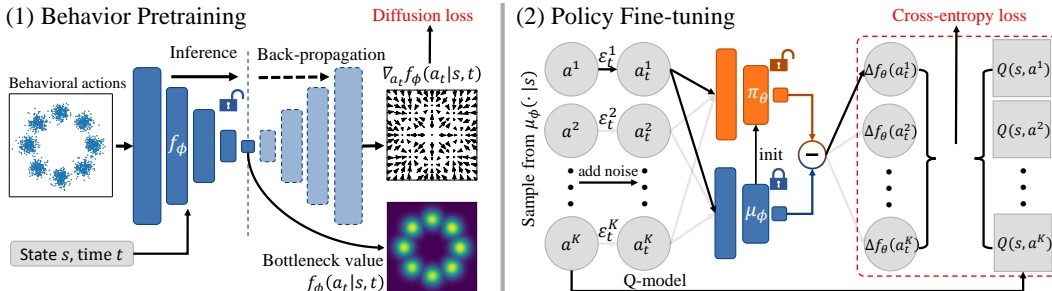

Figure 2: Algorithm overview. **Left:** In behavior pretraining, the diffusion behavior model is represented as the derivative of a scalar neural network with respect to action inputs. The scalar outputs of the network can later be utilized to estimate behavior density. **Right:** In policy fine-tuning, we predict the optimality of actions in a contrastive manner among $K$ candidates. The prediction logit for each action is the density gap between the learned policy model and the frozen behavior model. We use cross-entropy loss to align prediction logits $\triangle f_\theta := f_\theta^\pi - f_\theta^\mu$ with dataset Q-labels.

## 3 Method

We decompose the policy optimization problem into two stages: behavior pretraining (Sec. 3.1) and policy alignment (Sec. 3.2).

### 3.1 Bottleneck Diffusion Models for Efficient Behavior Density Estimation

Recent advances in alignment techniques cannot be readily applied to continuous control tasks. Their successful applications in LLM fine-tuning require two essential prerequisites:

1. A powerful foundation model capable of capturing diverse behaviors within datasets.
2. A tractable probability calculation method that allows direct density estimation (Eq. 4).

Language models primarily deal with discrete actions (tokens) defined by a vocabulary set $\mathcal{V}$, and thus employ Categorical models. This modeling method enables easy calculation of data probability through a softmax operation and is capable of representing *any* discrete distribution. In contrast, for continuous action space, direct density estimation is not so feasible. Diffusion policies only estimate the gradient field (a.k.a., score) of data density instead of the density value itself [49], making it impossible to directly apply LLM alignment techniques [41, 9, 3]. Conventional Gaussian policies have a tractable probability formulation but lack enough expressivity and multimodality needed to accurately model behavior datasets [52, 13, 5], and catastrophically fail in our initial experiments.

To address the above limitation of diffusion models, we propose a new diffusion modeling technique to enable direct density estimation. Normally, a conditional diffusion policy $\epsilon_\phi(a_t|s,t) : \mathcal{A} \times \mathcal{S} \times \mathbb{R} \to \mathbb{R}^{|\mathcal{A}|}$ maps noisy actions $a_t$ to predicted noises $\epsilon \in \mathbb{R}^{|\mathcal{A}|}$. In our approach, we redefine $\epsilon_\phi$ as the derivative of a scalar network $f_\phi(a_t|s,t) : \mathcal{A} \times \mathcal{S} \times \mathbb{R} \to \mathbb{R}$ with respect to input $a_t$:

$$\epsilon_\phi(a_t|s,t) := -\sigma_t \nabla_{a_t} f_\phi(a_t|s,t). \tag{9}$$

Given that $f_\phi$ is a parameterized network, its gradient computation can be conveniently performed by auto-differential libraries. The new training objective for $f_\phi$ can then be reformulated from Eq. 7:

$$\min_\phi \mathcal{L}_\mu(\phi) = \mathbb{E}_{t,\epsilon,(s,a)\sim\mathcal{D}^\mu} \left[ \|\sigma_t \nabla_{a_t} f_\phi(a_t|s,t) + \epsilon\|_2^2 \right]_{a_t = \alpha_t a + \sigma_t \epsilon}. \tag{10}$$

As noted by [49], with unlimited model capacity, the optimal solution for solving Eq. 10 is:

$$\epsilon^*(a_t|s,t) = -\sigma_t \nabla_{a_t} \log \mu_t(a_t|s,t) \implies f^*(a_t|s,t) = \log \mu_t(a_t|s,t) + C(s,t). \tag{11}$$

An illustration is provided in Figure 2 (left). Intuitively, our proposed modeling method first compresses the input action into a scalar value with one single dimension. Then, this bottleneck value is expanded back to $\mathbb{R}^{|\mathcal{A}|}$ through back-propagation. We thus refer to $f_\phi$ as *Bottleneck* Diffusion Models (BDMs).

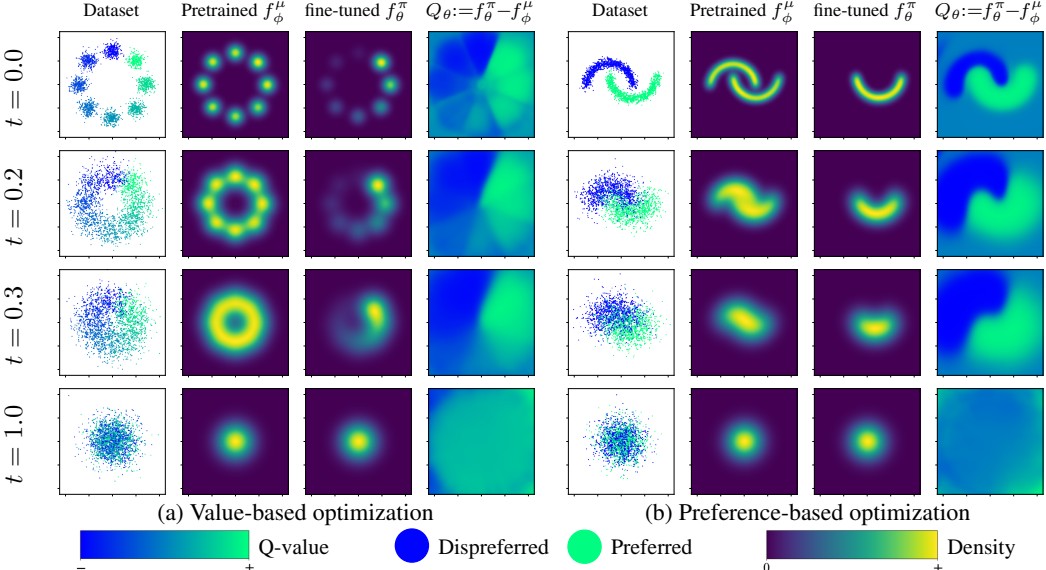

Figure 3: Experimental results of EDA in 2D bandit settings at different diffusion times. **Column 1:** Visualization of diversified behavior datasets. Each dot represents a two-dimensional behavioral action. Its color reflects the action's Q-value. **Column 2 & 3:** Density maps of the action distribution as estimated by the pretrained or fine-tuned BDM models. The density for low-Q-value actions has been effectively decreased after fine-tuning. **Column 4:** The predicted action Q-values, calculated by Eq. 13, align with dataset Q-values in Column 1. See appendix B for complete results.

The primary advantage of BDMs is their ability to efficiently estimate behavior densities in a single forward pass. Moreover, BDMs are fully compatible with existing diffusion-based codebases regarding training and sampling procedures, inheriting their key benefits such as training stability and model expressivity. BDMs can also be viewed as a diffused version of Energy-Based Models (EBMs, [8]). We refer interested readers to Appendix A for a detailed discussion.

## 3.2 Policy Optimization by Aligning Diffusion Behaviors with Q-functions

In this section, our goal is to learn a new policy $\pi_\theta \propto \mu_\phi e^Q$ by fine-tuning the previously pretrained behavior policy $\mu_\phi$ on a new dataset annotated by an existing Q-function $Q(s, a)$. We show that this policy optimization problem can actually be transformed into a simple classification task for predicting the optimal action among multiple candidates. We elaborate on our method below.

**Dataset construction.** For any state $s$, we draw $K > 1$ independent action samples $a^{1:K}$ from $\mu_\phi(\cdot|s)$. Assume we already have an $Q$-function $Q(s, a)$ that evaluates input state-action pairs in scalar values, our dataset is formed as $\mathcal{D}^f := \{s, a^{1:K}, Q(s, a^k)|_{k \in 1:K}\}$.

**Action optimality.** We first introduce a formal notion of action optimality. We draw from the control-as-probabilistic-inference framework [29] and define a random optimality variable $\mathcal{O}_K$, which is a one-hot vector of length $K$. The $k$-th index of $\mathcal{O}_K$ being 1 indicates that $a^k$ is the optimal action within $K$ action candidates $a^{1:K}$. We have

$$p(\mathcal{O}_K^k = 1 | s, a^{1:K}) = \frac{e^{Q(s, a^k)}}{\sum_{i=1}^K e^{Q(s, a^i)}}.$$

The optimality probability of a behavioral action $a$ is proportional to the exponential of its Q-value, aligning with the optimal policy definition $\pi^*(a|s) \propto \mu(a|s)e^{Q(s, a)/\beta}$.

**From policy optimization to action classification.** We construct a classification task by predicting the optimal action among $K$ candidates. This requires learning a Q-model termed $Q_\theta$ first. Drawing inspiration from DPO (Sec. 2.2), we parameterize $Q_\theta$ as the log probability ratio between $\pi_\theta$ and $\mu_\phi$:

$$Q_\theta(s, a) := \beta \log \frac{\pi_\theta(a|s)}{\mu_\phi(a|s)} + \beta \log Z(s),$$

This parameterization allows us to directly optimize $\pi_\theta$ during training because $\pi_\theta(\boldsymbol{a}|\boldsymbol{s}) = \frac{1}{Z(\boldsymbol{s})}\mu_\phi(\boldsymbol{a}|\boldsymbol{s})e^{Q_\theta(\boldsymbol{s},\boldsymbol{a})/\beta}$ constantly holds. Since $Q$-values in datasets define the probability of being the optimal action, the training objective can be derived by applying cross-entropy loss:

$$\max_\theta \mathcal{L}_\pi(\theta) = \mathbb{E}_{(\boldsymbol{s},\boldsymbol{a}^{1:K})\sim\mathcal{D}^f}\left[\sum_{k=1}^K \underbrace{\frac{e^{Q(\boldsymbol{s},\boldsymbol{a}^k)}}{\sum_{i=1}^K e^{Q(\boldsymbol{s},\boldsymbol{a}^i)}}}_{\text{optimality probability}} \log \underbrace{\frac{e^{\beta\log\frac{\pi_\theta(\boldsymbol{a}^k|\boldsymbol{s})}{\mu_\phi(\boldsymbol{a}^k|\boldsymbol{s})} + \beta\log Z(\boldsymbol{s})}}{\sum_{i=1}^K e^{\beta\log\frac{\pi_\theta(\boldsymbol{a}^i|\boldsymbol{s})}{\mu_\phi(\boldsymbol{a}^i|\boldsymbol{s})} + \beta\log Z(\boldsymbol{s})}}}_{\text{predicted probability}}\right]. \quad (12)$$

The unknown partition function $Z(\boldsymbol{s})$ automatically cancels out during division. $\beta$ is a hyperparameter that can be tuned to control how far $\pi_\theta$ deviates from $\mu_\phi$.

**Expanding to bottleneck diffusion behavior.** A reliable behavior density estimation of $\mu_\phi(\boldsymbol{a}|\boldsymbol{s})$ is critical and is a main challenge for applying Eq. 12. Our initial experiments tested with Gaussian policies drastically failed and even underperformed vanilla behavior cloning. This highlights the necessity of adopting a much more powerful generative policy, such as the BDM model (Sec. 3.1).

Diffusion policies define a series of distributions $\pi_t(\boldsymbol{a}_t|\boldsymbol{s},t) = \int \mathcal{N}(\boldsymbol{a}_t|\alpha_t\boldsymbol{a}_0, \sigma_t^2\boldsymbol{I})\pi_0(\boldsymbol{a}_0|\boldsymbol{s},t)\mathrm{d}\boldsymbol{a}_0$ at different timesteps $t \in [0,1]$, rather than just a single distribution $\pi = \pi_0$. Consequently, instead of directly predicting the optimal action given raw actions $\boldsymbol{a}^{1:K}$, we perturb all actions with $K$ independent Gaussian noises according to the diffusion forward process by applying $\boldsymbol{a}_t^k = \alpha_t\boldsymbol{a}^k + \sigma_t\boldsymbol{\epsilon}^k$. Then we predict action optimality given $K$ noisy action $\boldsymbol{a}_t^{1:K}$:

$$\begin{aligned}Q_\theta(\boldsymbol{s},\boldsymbol{a}_t,t) :=&\beta\log\frac{\pi_{t,\theta}(\boldsymbol{a}_t|\boldsymbol{s},t)}{\mu_{t,\phi}(\boldsymbol{a}_t|\boldsymbol{s},t)} + \beta\log Z(\boldsymbol{s})\\ =&\beta[f_\theta^\pi(\boldsymbol{a}_t|\boldsymbol{s},t) - f_\phi^\mu(\boldsymbol{a}_t|\boldsymbol{s},t)] + \beta[\log Z(\boldsymbol{s},t) - C^\pi(\boldsymbol{s},t) + C^\mu(\boldsymbol{s},t)],\end{aligned} \quad (13)$$

Similar to Eq. 12, all unknown terms above automatically cancel out in the training objective:

$$\max_\theta \mathcal{L}_f(\theta) = \mathbb{E}_{t,\boldsymbol{\epsilon}^{1:K},(\boldsymbol{s},\boldsymbol{a}^{1:K})\sim\mathcal{D}^f}\left[\sum_{k=1}^K \underbrace{\frac{e^{Q(\boldsymbol{s},\boldsymbol{a}^k)}}{\sum_{i=1}^K e^{Q(\boldsymbol{s},\boldsymbol{a}^i)}}}_{\text{optimality probability}} \log \underbrace{\frac{e^{\beta[f_\theta^\pi(\boldsymbol{a}_t^k|\boldsymbol{s},t) - f_\phi^\mu(\boldsymbol{a}_t^k|\boldsymbol{s},t)]}}{\sum_{i=1}^K e^{\beta[f_\theta^\pi(\boldsymbol{a}_t^i|\boldsymbol{s},t) - f_\phi^\mu(\boldsymbol{a}_t^i|\boldsymbol{s},t)]}}}_{\text{predicted probability on noisy actions}}\right]. \quad (14)$$

**Proposition 3.1.** *(Proof in Appendix C) Let $f_\theta^*$ be the optimal solution of Problem 14 and $\pi_{t,\theta}^* \propto e^{f_\theta^*}$ be the optimal diffusion policy. Assuming unlimited model capacity and data samples, we have the following results:*

*(a) **Optimality Guarantee.** At time $t = 0$, the learned policy $\pi_\theta^*$ converges to the optimal target policy.*

$$\pi_\theta^*(\boldsymbol{a}|\boldsymbol{s}) = \pi_{t=0,\theta}^*(\boldsymbol{a}|\boldsymbol{s}) \propto \mu_\phi(\boldsymbol{a}|\boldsymbol{s})e^{Q(\boldsymbol{s},\boldsymbol{a})/\beta}$$

*(b) **Diffusion Consistency.** At time $t > 0$, $\pi_{t>0,\theta}$ models the diffused distribution of $\pi_\theta^*$:*

$$\pi_{t,\theta}^*(\boldsymbol{a}|\boldsymbol{s},t) = \int \mathcal{N}(\boldsymbol{a}_t|\alpha_t\boldsymbol{a}, \sigma_t^2\boldsymbol{I})\pi_\theta^*(\boldsymbol{a}_0|\boldsymbol{s})\mathrm{d}\boldsymbol{a}_0$$

*asymptotically holds when $K \to \infty$ and $\beta = 1$, satisfying the definition of diffusion process (Eq. 6).*

**Fine-tuning Efficiency.** In practice, the policy and the behavior model share the same architecture, so we can initialize $\theta = \phi$ to fully exploit the generalization capabilities acquired during the pretraining phase. This fine-tuning technique allows us to perform policy optimization with an incredibly small amount of $Q$-labelled data (e.g., 10k samples) and optimization steps (e.g., 20K gradient steps). These are less than 5% of previous approaches (Sec. 5.2).

## 4 Related Work

### 4.1 Diffusion Modeling for Offline Continuous Control

Recent advancements in offline RL have identified diffusion models as an effective approach for behavior modeling [38, 16], which excels at representing complex and multimodal distributions

| Environment | Dataset | BCQ | CQL | IQL | DT | D-Diffuser | IDQL | Diffusion-QL | QGPO | BDM (Ours) |
|---|---|---|---|---|---|---|---|---|---|---|
| HalfCheetah | Medium-Expert | 64.7 | 91.6 | 86.7 | 86.8 | 90.6 | **95.9** | **96.8** | 93.5 | $93.2 \pm 1.2$ |
| HalfCheetah | Medium | 40.7 | 44.0 | 47.4 | 42.6 | 49.1 | 51.0 | 51.1 | 54.1 | $\mathbf{57.0 \pm 0.5}$ |
| HalfCheetah | Medium-Replay | 38.2 | 45.5 | 44.2 | 36.6 | 39.3 | 45.9 | 47.8 | 47.6 | $\mathbf{51.6 \pm 0.9}$ |
| Hopper | Medium-Expert | 100.9 | 105.4 | 91.5 | **107.6** | 111.8 | 108.6 | 111.1 | 108.0 | $104.9 \pm 7.4$ |
| Hopper | Medium | 54.5 | 58.5 | 66.3 | 67.6 | 79.3 | 65.4 | 90.5 | **98.0** | $\mathbf{98.4 \pm 3.9}$ |
| Hopper | Medium-Replay | 33.1 | 95.0 | 94.7 | 82.7 | **100.0** | 92.1 | **100.7** | 96.9 | $92.7 \pm 10.0$ |
| Walker2d | Medium-Expert | 57.5 | **108.8** | 109.6 | 108.1 | 108.8 | 112.7 | 110.1 | 110.7 | $\mathbf{111.1 \pm 0.7}$ |
| Walker2d | Medium | 53.1 | 72.5 | 78.3 | 74.0 | 82.5 | 82.5 | **87.0** | 86.0 | $\mathbf{87.4 \pm 1.1}$ |
| Walker2d | Medium-Replay | 15.0 | 77.2 | 73.9 | 66.6 | 75.0 | 85.1 | **95.5** | 84.4 | $89.2 \pm 5.5$ |
| **Average (D4RL Locomotion)** | | 50.9 | 77.6 | 76.9 | 74.7 | 81.8 | 82.1 | **88.0** | 86.6 | **87.3** |
| AntMaze | Umaze | 73.0 | 74.0 | 87.5 | 59.2 | - | **94.0** | 93.4 | 96.4 | $93.0 \pm 4.5$ |
| AntMaze | Umaze-Diverse | 61.0 | 84.0 | 62.2 | **53.0** | - | 80.2 | 66.2 | 74.4 | $\mathbf{81.0 \pm 7.4}$ |
| AntMaze | Medium-Play | 0.0 | 61.2 | 71.2 | 0.0 | - | 84.5 | 76.6 | 83.6 | $79.0 \pm 4.2$ |
| AntMaze | Medium-Diverse | 8.0 | 53.7 | 70.0 | 0.0 | - | 84.8 | 78.6 | 83.8 | $\mathbf{84.0 \pm 8.2}$ |
| **Average (D4RL Locomotion)** | | 35.5 | 68.2 | 72.7 | 28.1 | - | **85.9** | 78.7 | 84.6 | 84.3 |
| Kitchen | Complete | 8.1 | 43.8 | 62.5 | - | - | - | **84.0** | 62.8 | $81.5 \pm 7.3$ |
| Kitchen | Partial | 18.9 | 49.8 | 46.3 | - | 57.0 | - | 60.5 | 66.0 | $\mathbf{69.3 \pm 4.6}$ |
| Kitchen | Mixed | 8.1 | 51.0 | 51.0 | - | **65.0** | - | 62.6 | 45.5 | $\mathbf{65.3 \pm 2.2}$ |
| **Average (D4RL Locomotion)** | | 11.7 | 48.2 | 53.3 | - | - | - | 69.0 | 58.1 | **72.0** |
| **Average (D4RL Overall)** | | 39.7 | 69.7 | 71.4 | - | - | - | 82.1 | 80.7 | **83.7** |

Table 1: Evaluation results of D4RL benchmarks (normalized according to [10]). We report mean ± standard deviation of algorithm performance across 5 random seeds at the end of training. Numbers within 5 % of the maximum are highlighted.

compared with other modeling methods like Gaussians [54, 55, 24, 35, 53] or VAEs [11, 26, 12]. However, optimizing diffusion models can be a bit more tricky due to the unavailability of a tractable probability calculation method [5, 51]. Existing approaches include learning a separate guidance network to guide the sampling process during evaluation [21, 32], applying classifier-free guidance [1, 7], backpropagating sampled actions through the diffusion policy model to maximize Q-values [52, 22], distilling new policies from diffusion behaviors [13, 4], using rejection sampling to filter out behavioral actions with low Q-values [5, 15] and applying planning-based techniques [30, 36, 16]. Our proposed method differs from all previous work in that it directly fine-tunes the pretrained behavior to align with task-specific annotations, instead of learning a new downstream policy.

### 4.2 Preference-based Alignment in Offline Reinforcement Learning

Existing alignment algorithms are largely preference-based methods. Preference-based Reinforcement Learning (PbRL) assumes the reward function is unknown, and must be learned from data. Previous work usually applies inverse RL to learn a reward model first and then uses this reward model for standard RL training [19, 57, 28, 44, 34]. This separate learning of a reward model adds complexity to algorithm implementation and thus limits its application. To address this issue, recent work like OPPO [23], DPO [41], and CPL [17] respectively proposes methods to align Gaussian or Categorical policies directly with human preference. Despite their simplicity and effectiveness, these techniques require calculating model probability, and thus cannot be applied to diffusion policies. Existing diffusion alignment strategies [7, 56, 2, 51] are incompatible with mainstream PbRL methods. Our work effectively closes this gap by introducing bottleneck diffusion models. We also extend existing PbRL methods to align with continuous Q-functions instead of just binary preference data.

## 5 Experiments

We conduct experiments to answer the following questions:

- How does EDA perform compared with other baselines in standard benchmarks? (Sec. 5.1)

- Is the alignment stage data-efficient and training-efficient? How many training steps and annotated data does EDA require for aligning pretrained behavior models? (Sec. 5.2)

- Does value-based alignment outperform preference-based alignment? (Sec. 5.3)

- How do contrastive action number $K$ and other choices affect the performance? (Sec. 5.4)

## 5.1 D4RL Evaluation

**Benchmark.** In Table 1, we evaluate the performance of EDA in D4RL benchmarks [10]. All evaluation tasks have a continuous action space and can be broadly categorized into three types: `MuJoCo locomotion` are tasks for controlling legged robots to move forward, where datasets are generated by a variety of policies, including expert, medium, and mixed levels. `Antmaze` is about an ant robot navigating itself in a maze and requires both low-level control and high-level navigation. `FrankaKitchen` are manipulation tasks containing real-world datasets. It is critical to faithfully imitate human behaviors in these tasks.

**Experimental setup.** Throughout our experiments, we set the contrastive action number $K = 16$. For each task, we train an action evaluation model $Q_\psi(s, a)$ for annotating behavioral data during the fine-tuning stage. Implicit Q-learning [25] is employed for training $Q_\psi$ due to its simplicity and orthogonality to policy training. We compare with other critic training methods in Figure 4. The rest of the implementation details are in Appendix D.

**Baselines.** We mainly consider diffusion-based RL methods with various optimization techniques. `Decision Diffuser` [1] employs classifier-free guidance for optimizing behavior models. `QGPO` employs energy guidance. `IDQL` [15] does not optimize the behavior policy and simply uses rejection sampling during evaluation. `Diffusion-QL` [52] has no explicit behavior model, but instead adopts a diffusion regularization loss. We also reference well-studied conventional algorithms for different classes of generative policies. `BCQ` [11] features a VAE-based policy. `DT` [6] has a transformer-based architecture. `CQL` [27] and `IQL` [25] targets Gaussian policies.

**Result analysis.** From table 1, we find that EDA surpasses all referenced baselines regarding overall performance and provides a competitive number in each D4RL task. To ascertain whether this improvement stems from the alignment algorithm rather than from a superior critic model or policy class, we conduct additional controlled experiments. As outlined in Figure 4, we evaluate three variants of EDA using different Q-learning approaches and compare these against both diffusion and Gaussian baselines. The experimental results highlight the superiority of diffusion policies and further validate the effectiveness of our proposed method.

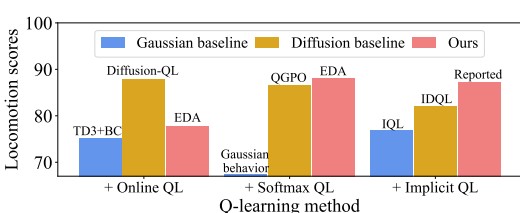

Figure 4: Average performance of EDA combined with different Q-learning methods in Locomotion tasks.

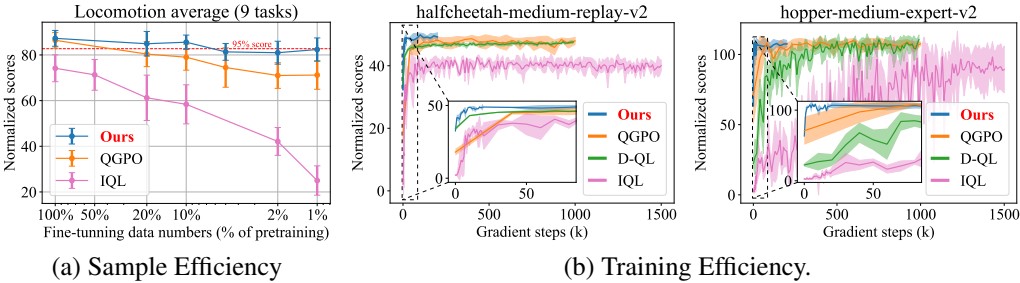

(a) Sample Efficiency          (b) Training Efficiency.

Figure 5: Aligning pretrained diffusion behaviors with task Q-functions is fast and data-efficient.

## 5.2 Fine-tuning Efficiency

The success of the pretraining, fine-tuning paradigm in language models is largely due to its high fine-tuning efficiency, allowing pretrained models to adapt quickly to various downstream tasks. Similarly, we aim to explore data efficiency and training efficiency during EDA's fine-tuning phase.

To investigate EDA's data efficiency, we reduce the training data used for aligning with pretrained Q-functions by randomly excluding a portion of the available dataset (Figure 5 (a)). We compare this with IQL, which uses the same Q-model as EDA but extracts a Gaussian policy via weighted regression. We also compare with QGPO, which shares our pretrained diffusion behavior models

but employs a separate guidance network to augment the behavior model during evaluation, instead of directly fine-tuning the pretrained policy. Experimental results reveal that EDA is significantly more data-efficient than the QGPO and IQL baselines. Notably, EDA maintains about 95% of its performance in locomotion tasks when using only 1% of the Q-labeled data for policy fine-tuning. This even surpasses several baselines that use the full dataset for policy training.

In Figure 5 (b), we plot EDA's performance throughout the fine-tuning phase. EDA rapidly converges in roughly 20K gradient steps, a negligible count compared to the typical 1M steps used for behavior pretraining. Note that behavior modeling and task-oriented Q-definition are largely orthogonal in offline RL. The high fine-tuning efficiency of EDA demonstrates the vast potential of pretraining on large-scale diversified behavior data and quickly adapting to individual downstream tasks.

### 5.3 Value Optimization v.s. Preference Optimization

A significant difference between EDA and existing preference-based RL methods such as DPO is that EDA is tailored for alignment with scalar Q-values instead of just preference data.

For preference data without explicit Q-labels, we can similarly derive an alignment loss:

$$\max_\theta \mathcal{L}_f^{\text{pref}}(\theta) = \mathbb{E}_{t,\boldsymbol{\epsilon}^{1:K},(\boldsymbol{s},\boldsymbol{a}^{1:K})\sim\mathcal{D}^f}\left[\log\frac{e^{\beta[f_\theta^\pi(\boldsymbol{a}_t^w|\boldsymbol{s},t)-f_\phi^\mu(\boldsymbol{a}_t^w|\boldsymbol{s},t)]}}{\sum_{i=1}^K e^{\beta[f_\theta^\pi(\boldsymbol{a}_t^i|\boldsymbol{s},t)-f_\phi^\mu(\boldsymbol{a}_t^i|\boldsymbol{s},t)]}}\right]. \tag{15}$$

Here $\boldsymbol{a}^w$ represents the most preferred action among $\boldsymbol{a}^{1:K}$. In practice, we select $\boldsymbol{a}^w$ as the action with the highest Q-value but abandon the absolute number. We'd like to note that when $K = 2$, the above objective becomes exactly the DPO objective (Eq. 4) in preference learning:

$$\max_\theta \mathcal{L}_f^{\text{DPO}}(\theta) = \mathbb{E}_{t,\boldsymbol{\epsilon}^{\{w,l\}},\boldsymbol{s},\boldsymbol{a}^w\succ\boldsymbol{a}^l}\log\sigma\left[\beta[f_\theta^\pi(\boldsymbol{a}_t^w|\boldsymbol{s},t)-f_\phi^\mu(\boldsymbol{a}_t^w|\boldsymbol{s},t)]-\beta[f_\theta^\pi(\boldsymbol{a}_t^l|\boldsymbol{s},t)-f_\phi^\mu(\boldsymbol{a}_t^l|\boldsymbol{s},t)]\right] \tag{16}$$

We ablate different choices of $K$ and compare our proposed value-based alignment with preference methods in 9 D4RL Locomotion tasks (Figure 6). Results show that EDA generally outperforms preference-based alignment approaches. We attribute this improvement to its ability to exploit the Q-value information provided by the pretrained Q-model. Besides, we notice the performance gap between the two methods becomes larger as $K$ increases. This is expected because preference-based optimization greedily follows a single action with the highest Q-value. However, as more action candidates are sampled from the behavior model, the final selected action will have a higher probability of being out-of-behavior-distribution data. This further hurts performance. In contrast, EDA is a softer version of preference learning. This leads to greater tolerance for $K$.

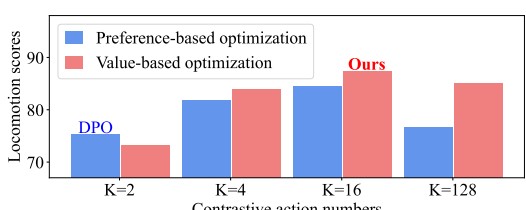

Figure 6: Ablation of action numbers $K$ and optimization methods.

### 5.4 Ablation Studies

We study the impact of varying temperature $\beta$ on algorithm performance in Appendix E. We also illustratively compare our proposed diffusion models with other generative models for behavior modeling in 2D settings in Appendix A.

## 6 Conclusion

We propose Efficient Diffusion Alignment (EDA) for solving offline continuous control tasks. EDA allows leveraging abundant and diverse behavior data to enhance generalization through behavior pretraining and enables rapid adaptation to downstream tasks using minimal annotations. Our experimental results show that EDA exceeds numerous baseline methods in D4RL tasks. It also demonstrates high sample efficiency during the fine-tuning stage. This indicates its vast potential in learning from large-scale behavior datasets and efficiently adapting to individual downstream tasks.

## Acknowledgments and Disclosure of Funding

We would like to thank Chengyang Ying and Cheng Lu for discussion. This work was supported by NSFC Projects (Nos. 62350080, 92248303, 92370124, 62276149, 62061136001), BNRist (BNR2022RC01006), Tsinghua Institute for Guo Qiang, and the High Performance Computing Center, Tsinghua University. J. Zhu was also supported by the XPlorer Prize.

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

# A    Comparing Bottleneck Diffusion Models with Energy-Based Models

Our proposed Bottleneck Diffusion Models (BDMs) can be viewed as a diffused variant of Energy-Based Models (EBMs, [8]). Both methods aim to model data distribution's unnormalized log probability:

$$p_\theta(\mathbf{x}) \propto e^{-E_\theta(\mathbf{x})}.$$

Despite their conceptual similarity, the sampling and training approaches differ between diffusion models and EBMs. A prevalent sampling method for EBMs is Langevin MCMC [14], which employs the energy gradient $\nabla_\mathbf{x} E_\theta(\mathbf{x})$ to progressively transform Gaussian noise into data samples. Langevin MCMC generally necessitates hundreds or thousands of iterative steps to achieve convergence, a significantly higher number compared with 15-50 steps required by diffusion models (Figure A).

Furthermore, the maximum-likelihood training of EBMs (e.g., Contrastive Divergence [18]) is more computationally expensive, as it involves online data sampling from the model during the training process [48]. To avoid online data sampling, subsequent research [50, 47] has shifted away from directly modeling $E_\theta(\mathbf{x})$ towards developing score-based models defined as $s_\theta(\mathbf{x}) := -\nabla_\mathbf{x} E_\theta(\mathbf{x})$. The score-matching objectives utilized in training these score-based models have subsequently been adapted for training diffusion models [49]. Our work is inspired by some prior work [42, 47, 31] that employed such score-matching objectives for training EBMs.

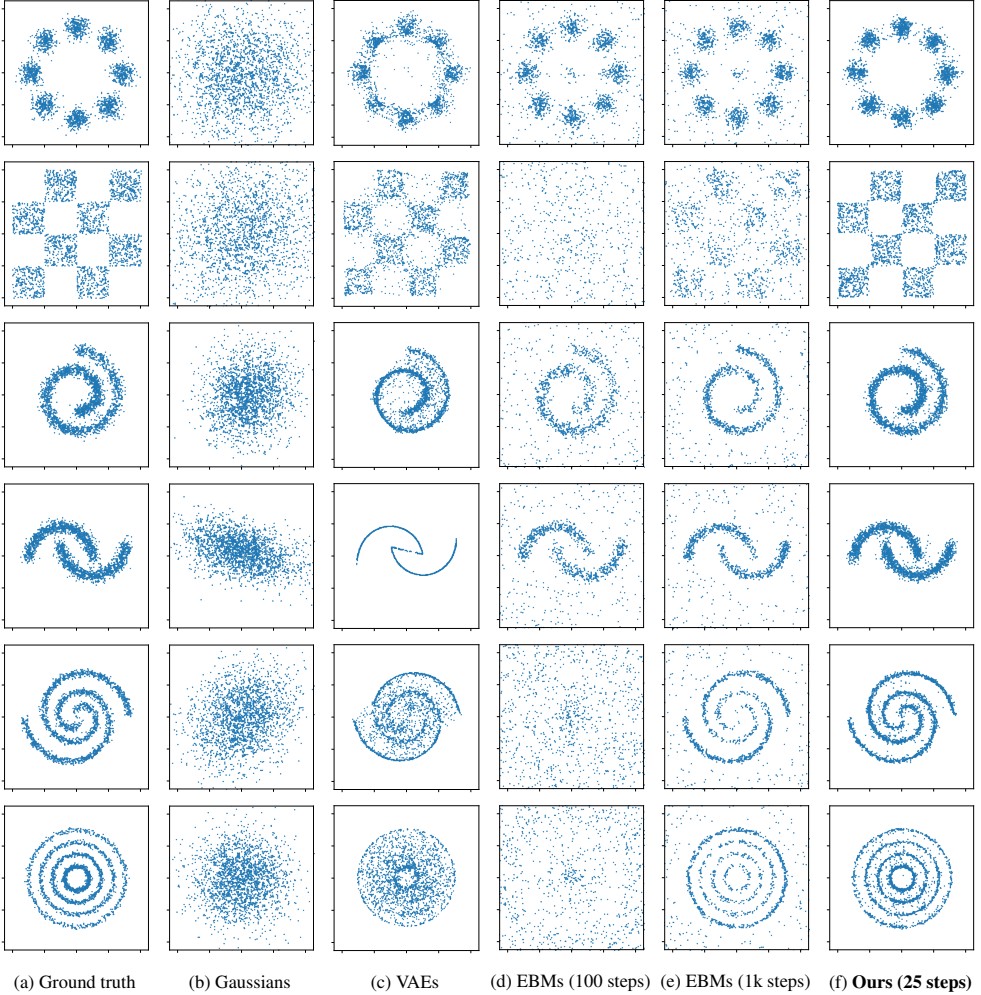

(a) Ground truth    (b) Gaussians    (c) VAEs    (d) EBMs (100 steps)  (e) EBMs (1k steps)  (f) **Ours (25 steps)**

Figure 7: Comparison of various generative modeling methods in 2D modeling and sampling.

# B Complete 2D-bandit Experiment Results

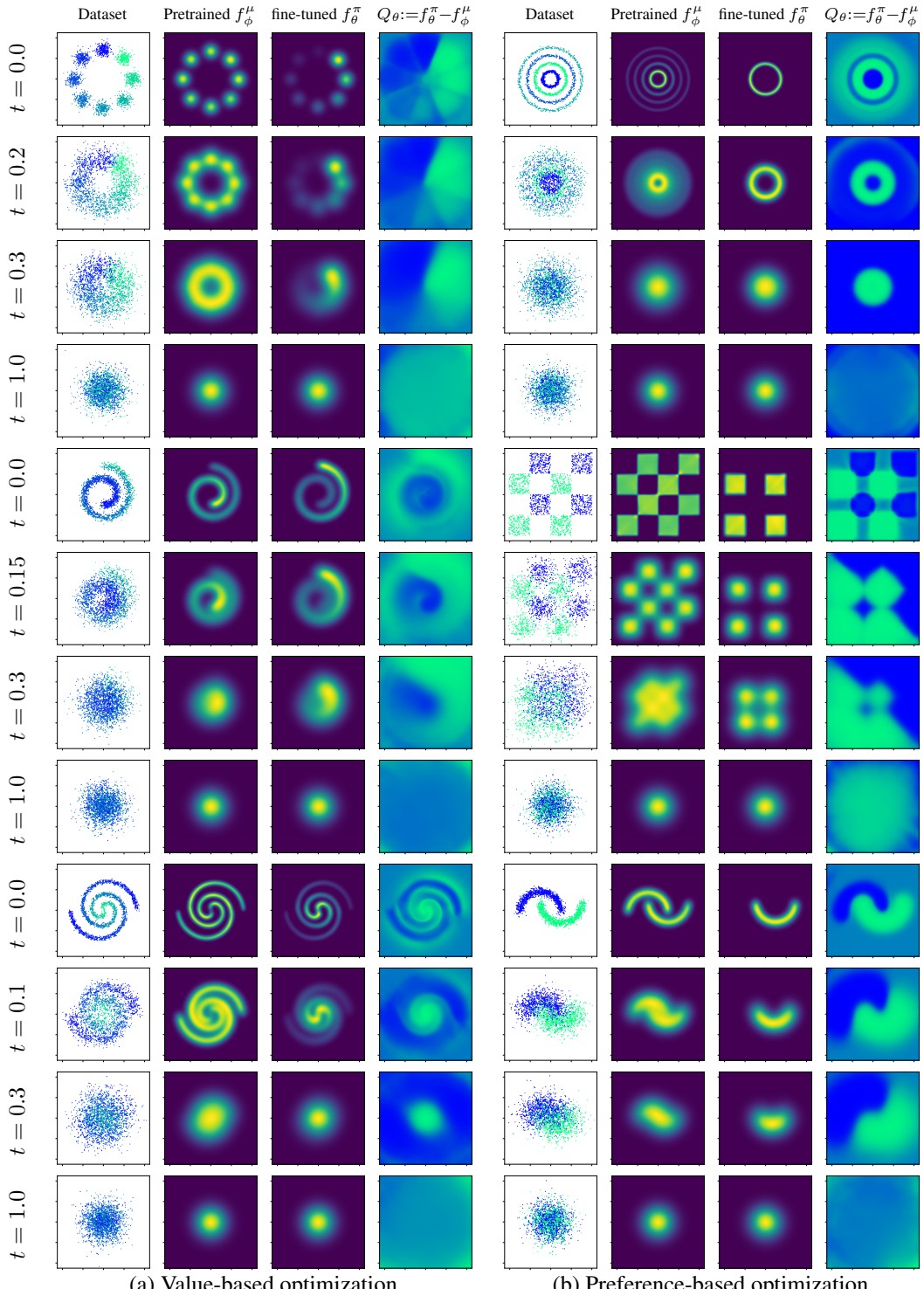

(a) Value-based optimization      (b) Preference-based optimization

Figure 8: Illustration of EDA's performance in 2D bandit settings at different diffusion times.

# C  Theoretical Analysis

In this section, we present the theoretical proof for Proposition 3.1.

Our proof is based on the theoretical framework of Contrastive Energy Prediction (CEP) for diffusion energy guidance [32]. For the ease of readers, we incorporate the relevant theories from their work as lemmas below.

**Lemma C.1.** *Let $\hat{Q}(s, a) : \mathcal{A} \times \mathcal{S} \to \mathbb{R}$ be a scalar function approximator. Consider the optimization problem*

$$\max_{\hat{Q}} \mathbb{E}_{\mu(s) \prod_{i=1}^{K} \mu(a^i|s)} \left[ \sum_{k=1}^{K} \frac{e^{Q(s,a^k)}}{\sum_{i=1}^{K} e^{Q(s,a^i)}} \log \frac{e^{\hat{Q}(s,a^k)}}{\sum_{i=1}^{K} e^{\hat{Q}(s,a^i)}} \right]. \tag{17}$$

*The solution for Problem 17 satisfies*

$$\hat{Q}^*(s, a) = Q(s, a) + C(s),$$

*where $C(s)$ can be arbitrary scalar functions conditioned on $s$.*

*Proof.* The proof is quite straightforward. Consider two discrete distributions

$$P := \left\{ \frac{e^{Q(s,a^1)}}{\sum_{i=1}^{K} e^{Q(s,a^i)}}, \frac{e^{Q(s,a^2)}}{\sum_{i=1}^{K} e^{Q(s,a^2)}} \cdots, \frac{e^{Q(s,a^K)}}{\sum_{i=1}^{K} e^{Q(s,a^i)}} \right\}$$

$$\hat{P} := \left\{ \frac{e^{\hat{Q}(s,a^1)}}{\sum_{i=1}^{K} e^{\hat{Q}(s,a^i)}}, \frac{e^{\hat{Q}(s,a^2)}}{\sum_{i=1}^{K} e^{\hat{Q}(s,a^i)}} \cdots, \frac{e^{\hat{Q}(s,a^K)}}{\sum_{i=1}^{K} e^{\hat{Q}(s,a^i)}} \right\}$$

For any $s$ and any $a^{1:K}$, we have

$$\mathbb{E}_{\mu(s) \prod_{i=1}^{K} \mu(a^i|s)} \left[ \sum_{k=1}^{K} \frac{e^{Q(s,a^k)}}{\sum_{i=1}^{K} e^{Q(s,a^i)}} \log \frac{e^{\hat{Q}(s,a^k)}}{\sum_{i=1}^{K} e^{\hat{Q}(s,a^i)}} \right]$$

$$= \mathbb{E}_{\mu(s) \prod_{i=1}^{K} \mu(a^i|s)} - D_{\text{KL}}(P||\hat{P}) - H(P)$$

$$\leq \mathbb{E}_{\mu(s) \prod_{i=1}^{K} \mu(a^i|s)} - H(P)$$

According to the properties of KL divergence, the equality holds if and only if $P = \hat{P}$ for any $s$ and $a^{1:K}$. This implies that

$$\hat{Q}^*(s, a) = Q(s, a) + C(s),$$

constantly holds. ∎

**Lemma C.2.** *Let $\hat{Q}(s, a_t, t) : \mathcal{A} \times \mathcal{S} \times \mathbb{R} \to \mathbb{R}$ be a scalar function approximator. $p(a_t|a, t)$ is any conditional transition probability. Consider the optimization problem*

$$\max_{\hat{Q}} \mathbb{E}_{\mu(s) \prod_{i=1}^{K} \mu(a^i|s)p(a_t^i|a^i,t)} \left[ \sum_{k=1}^{K} e^{Q(s,a^k)} \log \frac{e^{\hat{Q}(s,a_t^k,t)}}{\sum_{i=1}^{K} e^{\hat{Q}(s,a_t^i,t)}} \right]. \tag{18}$$

*The solution for Problem 18 satisfies*

$$\hat{Q}^*(s, a_t, t) = \log \mathbb{E}_{\mu_t(a|a_t,s,t)} e^{Q(s,a)} + C(s),$$

*where $\mu_t(a|a_t, s, t) = \mu(a|s)p(a_t|a, t)/\mu_t(a_t|s, t)$ is the posterior action distribution, $C(s)$ can be arbitrary scalar functions conditioned on $s$.*

*Proof.*

$$\mathbb{E}_{\mu(\boldsymbol{s}) \prod_{i=1}^{K} \mu(\boldsymbol{a}^i|\boldsymbol{s})p(\boldsymbol{a}_t^i|\boldsymbol{a}^i,t)} \left[ \sum_{k=1}^{K} e^{Q(\boldsymbol{s},\boldsymbol{a}^k)} \log \frac{e^{\hat{Q}(\boldsymbol{s},\boldsymbol{a}_t^k,t)}}{\sum_{i=1}^{K} e^{\hat{Q}(\boldsymbol{s},\boldsymbol{a}_t^i,t)}} \right]$$

$$= \mathbb{E}_{\mu(\boldsymbol{s}) \prod_{i=1}^{K} \mu_t(\boldsymbol{a}_t^i|\boldsymbol{s})\mu_t(\boldsymbol{a}^i|\boldsymbol{a}_t^i,\boldsymbol{s},t)} \left[ \sum_{k=1}^{K} e^{Q(\boldsymbol{s},\boldsymbol{a}^k)} \log \frac{e^{\hat{Q}(\boldsymbol{s},\boldsymbol{a}_t^k,t)}}{\sum_{i=1}^{K} e^{\hat{Q}(\boldsymbol{s},\boldsymbol{a}_t^i,t)}} \right].$$

$$= \mathbb{E}_{\mu(\boldsymbol{s}) \prod_{i=1}^{K} \mu_t(\boldsymbol{a}_t^i|\boldsymbol{s})} \left[ \sum_{k=1}^{K} \mathbb{E}_{\mu_t(\boldsymbol{a}^k|\boldsymbol{a}_t^k,\boldsymbol{s},t)} e^{Q(\boldsymbol{s},\boldsymbol{a}^k)} \log \frac{e^{\hat{Q}(\boldsymbol{s},\boldsymbol{a}_t^k,t)}}{\sum_{i=1}^{K} e^{\hat{Q}(\boldsymbol{s},\boldsymbol{a}_t^i,t)}} \right].$$

$$= \mathbb{E}_{\mu(\boldsymbol{s}) \prod_{i=1}^{K} \mu_t'(\boldsymbol{a}_t^i|\boldsymbol{s})} \left[ \sum_{k=1}^{K} \frac{\mathbb{E}_{\mu_t(\boldsymbol{a}^k|\boldsymbol{a}_t^k,\boldsymbol{s},t)} e^{Q(\boldsymbol{s},\boldsymbol{a}^k)}}{\sum_{i=1}^{K} \mathbb{E}_{\mu_t(\boldsymbol{a}^i|\boldsymbol{a}_t^i,\boldsymbol{s},t)} e^{Q(\boldsymbol{s},\boldsymbol{a}^i)}} \log \frac{e^{\hat{Q}(\boldsymbol{s},\boldsymbol{a}_t^k,t)}}{\sum_{i=1}^{K} e^{\hat{Q}(\boldsymbol{s},\boldsymbol{a}_t^i,t)}} \right].$$

According to Lemma C.1, for any state $\boldsymbol{a}$ and any diffused action $\boldsymbol{a}_t$ at time $t$, the optimal solution satisfies

$$\hat{Q}^*(\boldsymbol{s}, \boldsymbol{a}_t, t) = \log \mathbb{E}_{\mu_t(\boldsymbol{a}|\boldsymbol{a}_t,\boldsymbol{s},t)} e^{Q(\boldsymbol{s},\boldsymbol{a})} + C(\boldsymbol{s}).$$

$\square$

**Lemma C.3.** *Consider the behavior distribution $\mu(\boldsymbol{a}|\boldsymbol{s})$ and the policy distribution $\pi^*(\boldsymbol{a}|\boldsymbol{s}) \propto \mu(\boldsymbol{a}|\boldsymbol{s})e^{Q(\boldsymbol{s},\boldsymbol{a})}$. Their diffused distribution at time $t$ are both defined by the forward diffusion process (Eq. 5). Let $p(\boldsymbol{a}_t|\boldsymbol{a},t) := \mathcal{N}(\boldsymbol{a}_t|\alpha_t\boldsymbol{a}, \sigma_t^2 \boldsymbol{I})$, such that*

$$\mu_t(\boldsymbol{a}_t|\boldsymbol{s}, t) = \int \mathcal{N}(\boldsymbol{a}_t|\alpha_t\boldsymbol{a}, \sigma_t^2 \boldsymbol{I})\mu(\boldsymbol{a}|\boldsymbol{s}, t)\mathrm{d}\boldsymbol{a},$$

*and*

$$\pi_t^*(\boldsymbol{a}_t|\boldsymbol{s}, t) = \int \mathcal{N}(\boldsymbol{a}_t|\alpha_t\boldsymbol{a}, \sigma_t^2 \boldsymbol{I})\pi^*(\boldsymbol{a}|\boldsymbol{s}, t)\mathrm{d}\boldsymbol{a}.$$

Then the relationship between $\pi_t^*$ and $\mu_t$ can be derived as

$$\pi_t^*(\boldsymbol{a}_t|\boldsymbol{s}, t) \propto \mu_t(\boldsymbol{a}_t|\boldsymbol{s}, t)e^{Q_t(\boldsymbol{s},\boldsymbol{a}_t,t)},$$

where $Q_t(\boldsymbol{s}, \boldsymbol{a}_t, t) := \log \mathbb{E}_{\mu_t(\boldsymbol{a}|\boldsymbol{a}_t,\boldsymbol{s},t)} e^{Q(\boldsymbol{s},\boldsymbol{a})}$.

*Proof.* According to the definition, we have

$$\pi_t^*(\boldsymbol{a}|\boldsymbol{s}) = \int \mathcal{N}(\boldsymbol{a}_t|\alpha_t\boldsymbol{a}, \sigma_t^2 \boldsymbol{I})\pi^*(\boldsymbol{a}|\boldsymbol{s}, t)\mathrm{d}\boldsymbol{a}$$

$$= \int \mathcal{N}(\boldsymbol{a}_t|\alpha_t\boldsymbol{a}, \sigma_t^2 \boldsymbol{I})\mu(\boldsymbol{a}|\boldsymbol{s})\frac{e^{Q(\boldsymbol{s},\boldsymbol{a})}}{Z(\boldsymbol{s})}\mathrm{d}\boldsymbol{a}$$

$$= \int p(\boldsymbol{a}_t|\boldsymbol{a}, t)\mu(\boldsymbol{a}|\boldsymbol{s})\frac{e^{Q(\boldsymbol{s},\boldsymbol{a})}}{Z(\boldsymbol{s})}\mathrm{d}\boldsymbol{a}$$

$$= \int \mu_t(\boldsymbol{a}|\boldsymbol{a}_t, \boldsymbol{s}, t)\mu_t(\boldsymbol{a}_t|\boldsymbol{s}, t)\frac{e^{Q(\boldsymbol{s},\boldsymbol{a})}}{Z(\boldsymbol{s})}\mathrm{d}\boldsymbol{a}$$

$$= \frac{1}{Z(\boldsymbol{s})}\mu_t(\boldsymbol{a}_t|\boldsymbol{s}, t) \int \mu_t(\boldsymbol{a}|\boldsymbol{a}_t, \boldsymbol{s}, t)e^{Q(\boldsymbol{s},\boldsymbol{a})}\mathrm{d}\boldsymbol{a}$$

$$\propto \mu_t(\boldsymbol{a}_t|\boldsymbol{s}, t)e^{Q_t(\boldsymbol{s},\boldsymbol{a}_t,t)}$$

$\square$

**Proposition C.4.** *Let $f_\theta^*$ be the optimal solution of Problem 14 and $\pi_{t,\theta}^* \propto e^{f_\theta^*}$ be the optimal diffusion policy. Assuming unlimited model capacity and data samples, we have the following results:*

*(a) Optimality Guarantee. At time $t = 0$, the learned policy $\pi_\theta^*$ converges to the optimal target policy.*

$$\pi_\theta^*(\boldsymbol{a}|\boldsymbol{s}) = \pi_{t=0,\theta}^*(\boldsymbol{a}|\boldsymbol{s}) \propto \mu_\phi(\boldsymbol{a}|\boldsymbol{s})e^{Q(\boldsymbol{s},\boldsymbol{a})/\beta}$$

**(b) Diffusion Consistency.** *At time $t > 0$, $\pi_{t>0,\theta}$ models the diffused distribution of $\pi_\theta^*$:*

$$\pi_{t,\theta}^*(\boldsymbol{a}|\boldsymbol{s},t) = \int \mathcal{N}(\boldsymbol{a}_t|\alpha_t\boldsymbol{a}, \sigma_t^2\boldsymbol{I})\pi_\theta^*(\boldsymbol{a}_0|\boldsymbol{s})\mathrm{d}\boldsymbol{a}_0$$

*asymptotically holds when $K \to \infty$ and $\beta = 1$, satisfying the definition of diffusion process (Eq. 6).*

*Proof.* We first rewrite Problem 14 below:

$$\max_\theta \mathcal{L}_f(\theta) = \mathbb{E}_{\mu(\boldsymbol{s})\prod_{i=1}^K \mu(\boldsymbol{a}^i|\boldsymbol{s})p(\boldsymbol{a}_t^i|\boldsymbol{a}^i,t)}\left[\sum_{k=1}^K \frac{e^{Q(\boldsymbol{s},\boldsymbol{a}^k)}}{\sum_{i=1}^K e^{Q(\boldsymbol{s},\boldsymbol{a}^i)}} \log \frac{e^{\beta[f_\theta^\pi(\boldsymbol{a}_t^k|\boldsymbol{s},t)-f_\phi^\mu(\boldsymbol{a}_t^k|\boldsymbol{s},t)]}}{\sum_{i=1}^K e^{\beta[f_\theta^\pi(\boldsymbol{a}_t^i|\boldsymbol{s},t)-f_\phi^\mu(\boldsymbol{a}_t^i|\boldsymbol{s},t)]}}\right].$$
(19)

**(a) Optimality Guarantee.** At time $t = 0$, we have $p(\boldsymbol{a}_t|\boldsymbol{a},t) = \mathcal{N}(\boldsymbol{a}_t|\alpha_t\boldsymbol{a}, \sigma_t^2\boldsymbol{I}) = \mathcal{N}(\boldsymbol{a}_t|\boldsymbol{a}, 0\boldsymbol{I})$ such that $\boldsymbol{a}_t = \boldsymbol{a}$.

Define $\hat{Q}^*(\boldsymbol{s},\boldsymbol{a}) := \beta[f_\theta^\pi(\boldsymbol{a}_t^k|\boldsymbol{s},t=0) - f_\phi^\mu(\boldsymbol{a}_t^k|\boldsymbol{s},t=0)]$. Since we assume unlimited model capacity for $f_\theta$, $\hat{Q}^*$ can be arbitrary scalar functions. Lemma C.1 can then be applied:

$$\begin{aligned}
\pi_\theta^*(\boldsymbol{a}|\boldsymbol{s}) &= \pi_{t=0,\theta}^*(\boldsymbol{a}|\boldsymbol{s}) \\
&\propto e^{f_\theta^\pi(\boldsymbol{a}_t^i|\boldsymbol{s},t=0)} \\
&= e^{f_\phi^\mu(\boldsymbol{a}_t^i|\boldsymbol{s},t=0)+\hat{Q}^*(\boldsymbol{s},\boldsymbol{a})/\beta} \\
&= e^{f_\phi^\mu(\boldsymbol{a}_t^i|\boldsymbol{s},t=0)}e^{[Q(\boldsymbol{s},\boldsymbol{a})+C(\boldsymbol{s})]/\beta} \\
&\propto \mu_\phi(\boldsymbol{a}|\boldsymbol{s})e^{Q(\boldsymbol{s},\boldsymbol{a})/\beta}
\end{aligned}$$

**(b) Diffusion Consistency.** When $K \to \infty$, we $\sum_{i=1}^K e^{Q(\boldsymbol{s},\boldsymbol{a}^i)} = \mathbb{E}_\mu(\boldsymbol{a}|\boldsymbol{s})e^{Q(\boldsymbol{s},\boldsymbol{a})}$ becomes constant and can be removed. Set $\beta = 1$, the optimization problem equation 14 becomes

$$\max_\theta \mathcal{L}_f(\theta) = \mathbb{E}_{\mu(\boldsymbol{s})\prod_{i=1}^K \mu(\boldsymbol{a}^i|\boldsymbol{s})p(\boldsymbol{a}_t^i|\boldsymbol{a}^i,t)}\left[\sum_{k=1}^K e^{Q(\boldsymbol{s},\boldsymbol{a}^k)} \log \frac{e^{[f_\theta^\pi(\boldsymbol{a}_t^k|\boldsymbol{s},t)-f_\phi^\mu(\boldsymbol{a}_t^k|\boldsymbol{s},t)]}}{\sum_{i=1}^K e^{[f_\theta^\pi(\boldsymbol{a}_t^i|\boldsymbol{s},t)-f_\phi^\mu(\boldsymbol{a}_t^i|\boldsymbol{s},t)]}}\right]. \quad (20)$$

We can then similarly apply Lemma C.2 and get

$$\log \frac{\pi_{t,\theta}^*(\boldsymbol{a}_t|\boldsymbol{s},t)}{\mu_t(\boldsymbol{a}_t|\boldsymbol{s},t)} = f_\theta^\pi(\boldsymbol{a}_t|\boldsymbol{s},t) - f_\phi^\mu(\boldsymbol{a}_t|\boldsymbol{s},t) = \log \mathbb{E}_{\mu_t(\boldsymbol{a}|\boldsymbol{a}_t,\boldsymbol{s},t)}e^{Q(\boldsymbol{s},\boldsymbol{a})} + Z(\boldsymbol{s})$$

$$\pi_{t,\theta}^*(\boldsymbol{a}_t|\boldsymbol{s},t) \propto \mu_t(\boldsymbol{a}_t|\boldsymbol{s},t)\mathbb{E}_{\mu_t(\boldsymbol{a}|\boldsymbol{a}_t,\boldsymbol{s},t)}e^{Q(\boldsymbol{s},\boldsymbol{a})}$$

According to Lemma C.3, we have

$$\pi_{t,\theta}^*(\boldsymbol{a}|\boldsymbol{s},t) = \int \mathcal{N}(\boldsymbol{a}_t|\alpha_t\boldsymbol{a}, \sigma_t^2\boldsymbol{I})\pi_\theta^*(\boldsymbol{a}_0|\boldsymbol{s})\mathrm{d}\boldsymbol{a}_0$$

$\square$

# D Implementation Details for D4RL Tasks

We use NVIDIA A40 GPU cards to run all experiments.

**Behavior pretraining.** For pretraining the bottleneck diffusion model, we extract a reward-free behavior dataset $\{s, a\}$ from $\mathcal{D}^\mu := \{s, a, r, s'\}$. We adopt the model architecture used by IDQL [15] and SRPO [4] but sum up the final $|\mathcal{A}|$-dimensional output to form a scalar network. The resulting model is basically a 6-layer MLP with residual connections, layer normalizations, and dropout regularization. We train the behavior network for 1M steps to ensure convergence. The batch size is 2048. The optimizer is Adam with a learning rate of 3e-4. We adopt default VPSDE [49] hyperparameters as the diffusion data perturbation method.

**Constructing alignment dataset.** First, we leverage $\mathcal{D}^\mu$ and use existing methods such as IQL [25] to learn a critic network that will later be utilized for data annotation. Then, for a random portion of state $s$ in $\mathcal{D}^\mu$, we leverage the pretrained behavior model to generate $K = 16$ action samples. The original action in $\mathcal{D}^\mu$ is thrown away. We use the critic model to annotate each state-action pair and store them together as the alignment dataset $\mathcal{D}^f := \{s, a^{1:K}, Q(s, a^k)|_{k \in 1:K}\}$. We use 2-layer MLPs with 256 hidden states. Critic training details is exactly the same with previous work [25]. Other critic learning methods used in ablation studies are also consistent with respective prior work [15, 32].

**Policy fine-tuning.** The policy network is initialized to be the behavior network. Throughout the training, we fix the behavior model and only optimize the policy model. Behavior weights are frozen. The optimizer is Adam and the learning rate is 5e-5. All policy models are trained for 200k gradient steps though we observe convergence at 20K steps in most tasks. We do not employ dropout regularization during fine-tuning because we find it harms performance. For the temperature coefficient, we sweep over $\beta \in \{0.1, 0.2, 0.3, 0.5, 0.8, 1.0, 2.0\}$ (Figure 9&10). Similarly to [11, 15, 5, 15], we find the performance can be improved by adopting a rejection sampling technique. We select the action with the highest Q-value among 4 candidates during evaluation. We do not use such techniques in experimental plots (Figure 5&11) to better reflect policy improvement. We use 20 test seeds in all experiments and report numbers at the end of training. We train all experiments independently with 5 seeds in our main experiments and 3-5 seeds in ablation studies.

# E    Additional Experiment Results

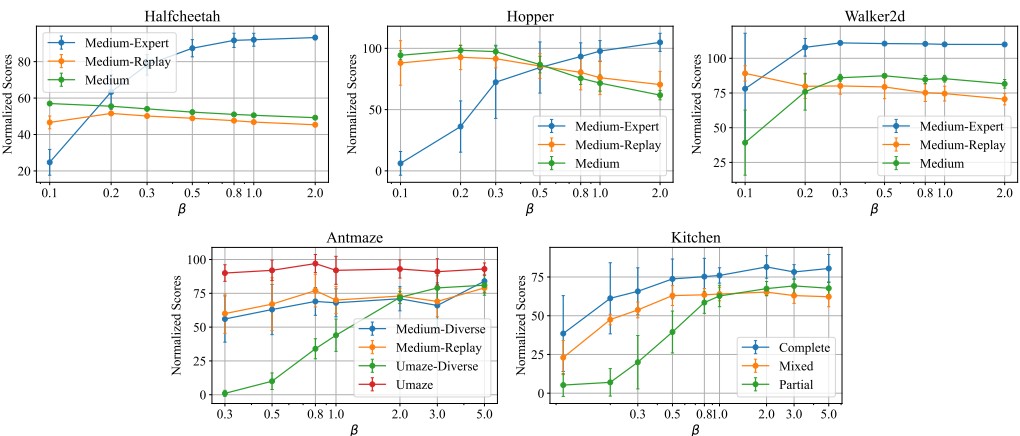

Figure 9: Ablation of the temperature coefficient $\beta$ in D4RL benchmarks.

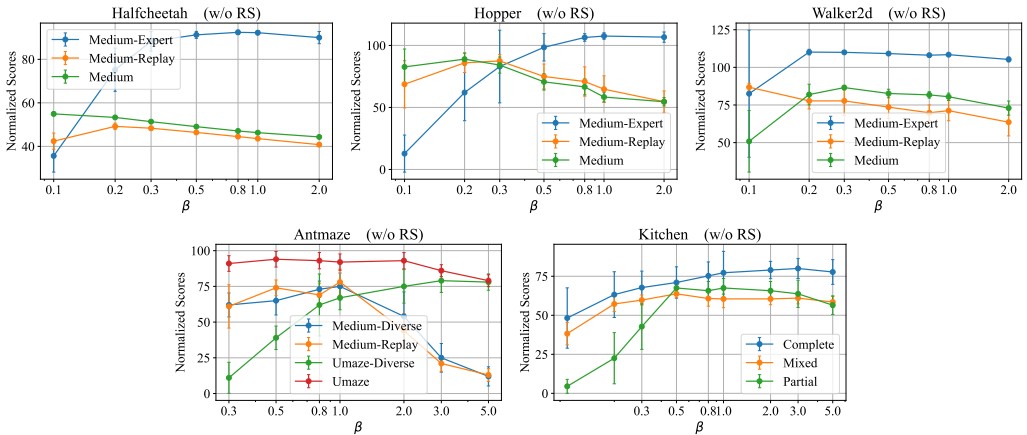

Figure 10: Ablation of the temperature coefficient $\beta$ without rejection sampling.

| | Medium-expert | Medium | Medium-Replay |
|---|---|---|---|
| Halfcheetah | 2.0 | 0.1 | 0.2 |
| Hopper | Medium-expert
2.0 | Medium
0.2 | Medium-Replay
0.2 |
| Walker2d | Medium-expert
0.3 | Medium
0.5 | Medium-Replay
0.1 |
| AntMaze | Umaze
0.5 | Umanze-diverse
5.0 | Medium (both)
1.0 |
| Kitchen | Complete
2.0 | Mixed
3.0 | Partial
3.0 |

Table 2: Temperature coefficient $\beta$ for every individual task.

# F Additional Training Curves

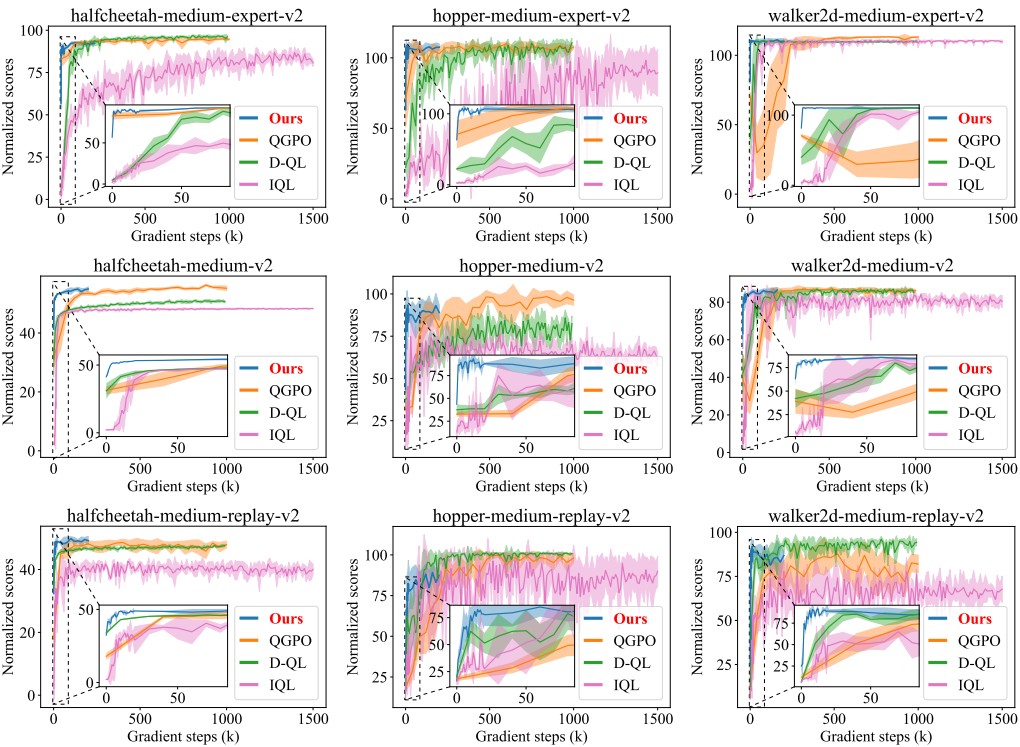

Figure 11: Training curves of EDA (ours) and several baselines.

## G    Reproducibility

To ensure that our work is reproducible, we submit the source code as supplementary material. The code will go open-source upon publication. Reported experimental numbers are averaged under multiple random seeds. We provide implementation details of our work in Appendix D.

## H    Limitations and Broader Impacts

**Limitations.** EDA employs a diffusion policy, which is a natural fit for continuous control problems. However, it cannot be directly applied in decision-making tasks with a discrete action space. Additionally, EDA places restrictions on the diffusion model design. The diffusion policy has to be the gradient of a scalar neural network with respect to action inputs to enable direct density estimation. Lastly,as task reward is human-defined and varies from task to task, achieving optimal performance with EDA in downstream tasks requires tailored network architecture design and hyperparameter tuning. How to ensure that the same set of algorithm parameters can be applicable across various tasks remains a topic worthy of further research.

**Broader Impacts.** EDA is a rather theoretical paper and all experiments are done in simulation environments. However, its further application could assist the deployment of highly optimized AI systems in critical real-world applications, such as autonomous driving or healthcare robots. This could have unintended consequences if the systems fail or produce incorrect results. Robust testing and validation are essential to mitigate these risks. There is also a risk that EDA could be misused for malicious purposes. For instance, optimized AI policies could be employed in autonomous systems for surveillance or other invasive activities without proper regulation and oversight.

