# OpenReview forum: "Aligning Diffusion Behaviors with Q-functions for Efficient Continuous Control"
_NeurIPS.cc/2024/Conference — NeurIPS 2024 poster_

### Official Review · Reviewer_FQV2 · 2024-06-28

**Soundness:** 3
**Presentation:** 3
**Contribution:** 3
**Rating:** 7
**Confidence:** 5

**Summary:**

This paper introduces Efficient Diffusion Alignment (EDA), which draws inspiration from the optimization paradigm of DPO to align the distribution of Diffusion policies with the optimal solutions of constrained policy optimization. Experimental results demonstrate the outstanding performance and fine-tuning efficiency of EDA.

**Strengths:**

- The paper is well-written and easy to follow.
- A standout aspect of the paper is the introduction of Bottleneck Diffusion Models (BDM), which utilize neural networks (NNs) to estimate the log-probability of noisy actions. In diffusion models, it is a classic paradigm to use NNs to estimate the score function (gradient of the log-probability), eliminating the need for gradient computations when solving reverse SDE/ODEs. This is particularly crucial in the application of large diffusion models, such as in image generation tasks. However, for smaller applications like action generation, the BDM approach, although requiring gradient computations on NNs during the denoising process, allows for flexible manipulation of likelihood. I believe this approach could lead to more diverse applications in future works beyond EDA.

**Weaknesses:**

- EDM shows only a marginal performance improvement compared to baselines in the experiments, being slightly better than DQL and QGPO.
- Using EDM in practice may be cumbersome. Firstly, manual preparation of an alignment dataset is required. Additionally, EDA seems to be sensitive to the temperature coefficient $\beta$, which is bound to the training process and cannot be adjusted during inference.

**Questions:**

Could the authors provide any insights on the selection of the number $K$ of actions in the state-action pairs within the alignment dataset? What is a sufficient value for $K$?

**Limitations:**

The paper only conducts experiments on benchmarks from D4RL. Including more real-world application benchmarks, such as Robomimic, may enhance the impact of the paper.

---

> ### Author Rebuttal · Authors · 2024-08-06
>
> # Official Response to Reviewer FQV2
> We completely agree with the reviewer's assessment of our paper, regarding both the praise and the pointed-out limitations. We thank the reviewer for providing valuable feedback and the reviewer's expertise in delivering such an insightful review, even though some deep motivations of our work were not extensively described in the text.
>
> **Q1:  Any insights on the selection of the number K of actions in the state-action pairs within the alignment dataset?"**
>
> **A1:**We would like to refer the reviewer to **Figure 8 in the paper, where we have already conducted ablation studies of K**.
> Our experience is that $K>8$ is already sufficient for reasonably good performance in D4RL tasks. For several environments like hopper,  $K>100$ will hurt performance (possibly because the Q-model is not good enough).
> A practical way to determine $K$ is to first try if rejection sampling can improve the performance of the pretrained behavior model: Selecting the best action with the highest Q-value among $K$ independently-sampled behavior actions.
>
> **Q2:  Using EDM in practice may be cumbersome. Firstly, manual preparation of an alignment dataset is required. Additionally, EDA seems to be sensitive to the temperature coefficient  β, which is bound to the training process and cannot be adjusted during inference."**
>
> **A2:** We agree these are critical limitations of the EDA method.
> 1. The $\beta$ needs to be predefined before training. This is indeed a bummer for EDA. There's currently a method (QGPO) where $\beta$, as a diffusion guidance scale, can be tuned during inference, which is realized by training an independent diffusion guidance network. However, QGPO cannot be initialized from pretrained diffusion behavior and thus has to be trained from scratch. We currently do not know how to combine the strengths of QGPO and EDA elegantly and think this could be a very important and fundamental topic. Its meaning is beyond the scope of offline RL but could benefit the entire research field such as LLM alignment/diffusion alignment.
> 2. How to remove the need for requiring multiple actions for one single state is another crucial topic that we are very interested in. Still, manual preparation of the dataset can be avoided by sampling actions during training in an online manner. However, this is a little bit computationally expensive and we find it not helpful for D4RL performance.
> 3. We think the sensitivity to $\beta$ is to some extent the problem of human-defined rewards. Since the reward signal is task-specific, it can be very difficult to apply the same hyperparameter to all tasks. We believe this is a common challenge for most RL methods (and perhaps a privilege for preference-based RL).
>
> **Q3: EDM shows only a marginal performance improvement compared to baselines in the experiments."**
>
> **A3:**
> 5. We'd like to stress that **D4RL is a highly competitive benchmark**. The **upper bound** score for D4RL is 100 and most baselines have achieved 80+. However, EDA still outperforms all baselines in the average performance of 16 tasks.
> 6. EDA demonstrates **obvious improvement in sample efficiency and convergence speed** (Figure 5). Given only 1% of data, EDA maintains 95% of performance while the best-performing baseline keeps only 80% of performance.  **We think this sample efficiency improvement is more critical"**. The high sample efficiency is an essential part for real-world application of the alignment algorithms.
>
>
> **Q4:  Including more real-world application benchmarks, such as Robomimic, may enhance the impact of the paper."**
>
> **A4:** We definitely agree with the reviewer.  As a matter of fact, we have considered applying EDA to multi-task,  general-purpose, and real-world agents just like Robomimic when we first came up with this idea. The main difficulty is that other benchmarks lack enough diffusion baselines to compare with. Still, this could be a very meaningful research direction.
>
> We hope the reviewer finds it acceptable that we list this as a future work and believe this does not harm the key contribution of the current article.

---

> > ### Comment · Reviewer_FQV2 · 2024-08-08
> >
> > Thank you for your detailed response. Firstly, I want to apologize for my typo in writing "EDA" as "EDM."
> >
> > I personally really like the experiments on sample efficiency. The fact that EDA shows almost no performance loss with only 1% of the data is quite surprising. I want to hear the authors' in-depth analysis and discussion of this result.  in the authors' view, what could be the main reasons behind this? Additionally, I strongly recommend that the authors consider adding a section in the paper for a detailed discussion on this topic. I believe this would bring many interesting insights.
> >
> > And, I still stand by my judgment. I keep the rating unchanged.

---

> ### Author Response · Authors · 2024-08-08
> **Additional Response**
>
> We thank the reviewer for the prompt reply and the interest in our work.
>
> We think the insight for being able to improve alignment efficiency here is that transforming $\mu$ into $\pi^*$ (alignment) is fundamentally much much easier than learning $\mu$ from scratch (pertaining). In alignment, we are simply trying to "suppress" some bad modes learned during pretraining instead of trying to find new meaningful modes.
>
> The theoretical explanation could be $KL(\mu\|\pi^*)$ << $KL(\mu\|\text{uniform dist})$. In high-dimensional data space, the actual meaningful data is actually very scarce. Imagine if you are trying to sample an image from Gaussian distribution, you wouldn't get a visually realistic image from even $10^{100}$ candidates. However, if you already have a pretrained image-generation model like stable-diffusion, you can easily get a good-looking one from 4-16 image samples.
>
> Why can EDA have high sample efficiency while other diffusion-based algorithms cannot? It is simply because EDA is completely initialized from the pretrained model. There are no network parameters that are required to be learned from scratch. Finetuning a model is much easier than learning a completely new one. Similar successes have already been proved by wide exploration in LLM alignment research.

---

> > ### Comment · Reviewer_FQV2 · 2024-08-08
> >
> > Thank you for the reply!
> >
> > So, does the statement "It is simply because EDA is completely initialized from the pretrained model." implies that during the behavior pretraining phase, EDA trains a behavior policy using the entire dataset and then uses 1% of the dataset during the fine-tuning phase? If that's the case, then the high sample efficiency of EDA is indeed easy to understand.

---

> ### Author Response · Authors · 2024-08-08
>
> Yes, though for the behavior pretraining dataset, reward labels are excluded. The diffusion model learns all kinds of behavior, regardless of whether they are good or bad.

---

### Official Review · Reviewer_y7D4 · 2024-07-11

**Soundness:** 3
**Presentation:** 2
**Contribution:** 3
**Rating:** 7
**Confidence:** 4

**Summary:**

This paper introduces Efficient Diffusion Alignment (EDA) for offline continuous control, combining the preference alignment theories with reinforcement learning. Specifically, EDA bridges the alignment finetuning by representing the diffusion score as a derivative of a scalar neural network with respect to actions, which allows direct density estimation. During fine tuning, DPO is used for policy improvement. In their experiments, EDA has demonstrated both strong performance and good sample efficiency in fine tuning on D4RL.

**Strengths:**

This is overall a novel and interesting paper. The way it connects preference-based optimisation with diffusion is simple yet effective, and allows efficient training of the diffusion-based RL policies. Besides, the authors have also provided theoretical justifications and toy examples for intuitive explanations to help the understanding. EDA also achieves a strong performance on the D4RL benchmark.

**Weaknesses:**

1/ The presentation has a certain room for improvement. The introduction made me very confused about why we want to do alignment, rather than directly optimising the policy with standard RL. This is never explained in the paper as well. After reading the whole paper, it is clearer to me. I do think this should be improved to make the paper an acceptance.

2/ Although EDA is simple by itself, the overall framework is quite complex as the training has been separated into multiple stages: pretraining of the diffusion policy, pretraining of the Q functions, and final alignment. This actually made the whole framework much more complex compared with the standard offline diffusion RL methods. It would be great if the authors could provide some explanations to the actual training time of EDA, compared with the baselines.

3/ Normally for pretraining and alignment, we are referring to training general purpose agents, and performing task-specific alignment. However, the current form of EDA doesn’t seem to support training general purpose agents that solve tasks with different action dimensions. Instead, it seems the current experiments are all conducted in a single-task manner, rather than multi-task settings. This actually hinders the contribution of the work.

4/ Considering the complexity of the framework, the proposed EDA doesn’t seem to provide much performance improvement compared with the baselines (83.7 overall return of EDA compared with 82.1 of DQL).

**Questions:**

1/ I believe this is a general framework that converts the policy optimization problem to alignment. Have you tried other alignment methods and how they perform?

2/ For a fixed environment but different offline datasets, have you tried pretraining on all data then performing alignment?

**Limitations:**

As discussed in the weaknesses, although the current method claimed to perform general pretraining, it seems to be performing single-task pretraining and alignment. Also, the overall pipeline is more complex and time consuming than the standard diffusion-based offline RL methods, and didn’t seem to provide much performance improvement. Nevertheless, I still think this is an interesting work, and I’ll vote for a weak acceptance.

---

> ### Author Rebuttal · Authors · 2024-08-06
>
> # Official Response to Reviewer y7D4
>
> We truly appreciate the reviewer for the valuable suggestions and comments. Concerns are mainly about the computational complexity of EDA and the experimentation/application of the algorithm.
>
> **Q1:  Could provide some explanations to the actual training time of EDA, compared with baselines."**
>
> **A1:**
> **Actual time tested on NVIDIA A40**:
>
> |Method|Behavior pretraining|Critic training|Policy Alignment|Dataset|Overall|
> |---|---|---|---|---|---|
> |Ours|3.4h (1M steps)|1.4h (1M steps)|0.07h (20k steps)|0.6h|5.5h|
> |IDQL|1.9h (1M steps)|1.4h (1M steps)|-|-|3.3h|
> |QGPO|1.9h (1M steps)|4.2h (1M steps)|3.2h (1M steps)|1.0h|10.3h|
> |DiffusionQL|-|4.1h (1M steps)|2.9h (1M steps)|-|7h|
>
> Overall the inference/training speed of BDM is less than 2 times of normal diffusion with the same model size. However, its advantage is
>
> **Density estimation**: **20+ times faster** than diffusion models. Similar to Gaussian and EBM models.
> **Alignment:** **Negligible computation** because EDA allows finetuning the model for very few steps instead of learning a brand-new model from scratch.
>
> **Q2:  The overall pipeline is more time consuming than the standard diffusion-based offline RL methods"**
>
> **A2:** We respectfully disagree with this comment. From **A1** we can see that in computational demand **EDA roughly matches other diffusion-based baselines**, and is somewhat more efficient because it has negligible policy alignment computation.
>
> **Q3:  EDA should support training general-purpose agents instead of conducting all experiments in a single-task manner. "**
>
> **A3:** We couldn't agree more with the reviewer on this comment.  This is actually the very first "hidden" motivation of our paper.  As a matter of fact, we considered applying EDA to multi-task settings and building general-purpose agents when we first came up with this idea, though we indeed faced some difficulties:
>
> 1. Since EDA is a pretty new idea, before moving on to large-scale/real-world experiments, we have to convince the research field as well as ourselves that it is actually effective and competitive. This means comparing with existing SOTA diffusion baselines in well-recognized continuous control tasks, where D4RL is the predominant benchmark. To our knowledge, there are **very few public benchmarks with a sufficient number of diffusion baselines** for meaningful comparisons besides D4RL.
>
> 2. We focus on the theoretical derivation and motivation of the EDA algorithm in this paper. **Applying EDA for general-purpose multi-task learning would  involve extensive engineering practices like data collection and task definition that are mostly orthogonal to the core focus of our original article.** This might divert the attention from the theoretical contributions of our paper, which we aim to emphasize.
>
> Overall, while we highly recognize the value of general-purpose experiments, we sincerely hope the reviewer finds it acceptable that we list this as a future work of our paper. **We believe this does not harm the key contribution of the current article.**
>
> **Q4:  Although EDA is simple by itself, the overall framework is quite complex (3 stages). This actually made the whole framework much more complex compared with the standard offline diffusion RL methods."**
>
> **A4:**
> Standard offline diffusion RL methods also require at least 2 stages of training (IQL/AWR/DiffusionQL). Some work also requires 3 stages (BCQ/BEAR/BRAC/QGPO)
>
> LLM training pipeline also requires three stages for training (Pretrain, SFT, alignment).
>
> **There are multiple stages of training because we need to handle distinctive data type and data amount in real-world application**
>
> For instance, in autonomous driving, we can collect all driving behaviors from drivers, which creates a feasible continuous action space and is helpful for learning a foundational end-to-end model. This requires large-scale pretraining.
>
> However, we may only have enough human resources to label all those very harmful actions that lead to car crashes and preferred actions that save drivers' lives. These data require efficient alignment.
>
> **Q5:  I believe this is a general framework that converts the policy optimization problem to alignment. Have you tried other alignment methods and how they perform?"**
>
> **A5:** We compare other classic preference-based LLM alignment methods including  DPO, SimPO, and IPO in our initial experiments. There's no very specific number but basically, they all perform similarly and DPO should be chosen given its simplicity and theoretical elegancy.
> Also, as our ablation results in Figure 6 have pointed out, we find that value-based alignment （EDA） outperforms preference-based approaches such as DPO.
>
> **Q6:  The improved performance seems marginal compared with the baselines."**
>
> **A6:**
> 1. We'd like to stress that **D4RL is a highly competitive benchmark**. The **upper bound** score for D4RL is 100 and  most baselines have achieved 80+. However EDA still outperforms all baselines in average performance of 16 tasks.
> 2. EDA demonstrates **obvious improvement in sample efficiency and convergence speed** (Figure 5). Given only 1% of data, EDA maintains 95% of performance while the best-performing baseline keeps only 80% of performance.  **We think this sample efficiency improvement is more critical"**. The high sample efficiency is an essential part of the real-world application of the alignment algorithms.
>
> **Q7:  For a fixed environment but different offline datasets, have you tried pretraining on all data and then performing alignment?"**
>
> **A7:** Unfortunately no, this is because for D4RL,  datasets for a single robot are largely overlapping/reused, for instance, the hopper-ME dataset is already a mixture of two datasets and hopper-M dataset is exactly its subset. Concatenating them together has not much empirical meaning.

---

> > ### Comment · Reviewer_y7D4 · 2024-08-09
> >
> > I thank the reviewer for the detailed responses and explanations. My concerns are well addressed and I will increase my rating to 7.

---

> ### Author Response · Authors · 2024-08-09
>
> We are glad that the reviewer is happy with our responses! We also appreciate the reviewer for the prompt and positive feedback.

---

### Official Review · Reviewer_jAma · 2024-07-13

**Soundness:** 3
**Presentation:** 3
**Contribution:** 3
**Rating:** 7
**Confidence:** 3

**Summary:**

This paper introduces Efficient Diffusion Alignment (EDA) for solving offline reinforcement learning problems. The approach involves first training a behavior cloning model using only state-action data, without reward information. Subsequently, the model is fine-tuned with rewards using DPO. During the reward-free training phase, a diffusion model is used as the policy network. Since the diffusion model doesn't provide the likelihood of the predicted action, it is modified to predict a scalar representing the density/energy of the action. The noise predicted by the model is computed by backpropagating to the input actions. Experiments are done on D4RL dataset, the paper shows that the proposed method can greatly improve the training efficiency.

**Strengths:**

- The paper is well-written and clearly explains the proposed method.
- The approach appears to be novel.

**Weaknesses:**

1. If I understand correctly, the proposed method needs backpropagation to compute the predicted noise and then another backpropagation to update the network. Would this require computing higher-order gradients?

2. If so, there is a lack of experiments and discussion about the efficiency and robustness of the method given the potential computational cost and sensitivity of computing higher-order gradients.

3. The method is only evaluated on the D4RL benchmark.

**Questions:**

1. Does Equation 10 require computing the gradient of the gradient for the proposed Bottleneck Diffusion Models? If so, this aspect should be experimentally evaluated and discussed.
2. Why was the diffusion model chosen as the policy network? Could other energy-based generative models, or even simpler models like GMM, be used to obtain data likelihood?
3. What are the benefits of using the proposed method over alternative methods that apply DPO on diffusion models in the image generation field, such as Diffusion-DPO?

Reference: Wallace, Bram, et al. "Diffusion model alignment using direct preference optimization." Proceedings of the IEEE/CVF Conference on Computer Vision and Pattern Recognition. 2024

**Limitations:**

- The paper lacks comparison with other energy-based generative models and simpler methods for obtaining data likelihood.
- The proposed method is only evaluated on the D4RL benchmark; additional benchmarks would strengthen the experimental validation.

---

> ### Author Rebuttal · Authors · 2024-08-05
>
> # Official Response to Reviewer jAma
> **Q1: Does EDA require backpropagation to compute the predicted noise and then another backpropagation to update the network? Would this involve computing higher-order gradients?**
>
> **A1:** Yes. The pretraining of BDM requires two backpropagation steps for a single training iteration.
>
> **However, we don't think it requires computing higher-order gradients** (such as the computationally expensive Jacobian Matrix).
>
> The BDM loss gradient simplifies to:
>
> $\frac{\partial L_\theta}{\partial \theta} = \frac{\partial }{\partial \theta} \|\frac{\partial }{\partial a_t}f_\theta(a_t,s, t)-\epsilon\|^2$
>
> Higher order gradient $\frac{\partial^2  f_\theta}{\partial  \theta \partial a_t}$ is computationally expensive.  However, such **Jacobian calculations can be automatically avoided by PyTorch** in practice.
>
> The basic logic is that: A computational graph can be constructed when calculating $\frac{\partial f_\theta}{\partial a_t}$. PyTorch will just regard $\frac{\partial f_\theta}{\partial a_t}$ as any normal neural network, except that it is 2-times larger.  **It is convenient to understand the proposed BDM models jus as a Normal network** with 2-times computation. (Figure 2 in paper)
>
> **Q2: Lack of experiments and discussion about the efficiency and robustness given the potential computational cost and sensitivity of computing higher-order gradients. "**
>
>
> **A2:**
> **computational cost： EDA roughly matches other diffusion-based baselines**, and is somewhat more efficient because it has negligible policy alignment computation. The inference/training speed of BDM is less than 2 times of normal diffusion with the same model size.
>
> | Method | Behavior pretraining | Critic training | Policy Alignment | Dataset | Overall |
> |---|---|---|---|---|---|
> | Ours|3.4h (1M steps) | 1.4h (1M steps) | 0.07h (20k steps) | 0.6h | 5.5h |
> | IDQL|1.9h (1M steps) | 1.4h (1M steps) | - | - | 3.3h |
> | QGPO | 1.9h (1M steps) | 4.2h (1M steps) | 3.2h (1M steps) | 1.0h | 10.3h |
> | DiffusionQL | - | 4.1h (1M steps) | 2.9h (1M steps) | - | 7h |
>
> **robustness：** including derivative in loss function has **already been verified by various research fields**. Application in diffusion modeling for image generation: [1] Application in Training PINN : [2]
>
> [1] Should EBMs model the energy or the score? Tim Salimans, Jonathan Ho
>
> [2] Scientific machine learning through physics–informed neural networks.
>
> **Q3: Could other energy-based generative models, or even simpler models like GMM, be used to obtain data likelihood?  Why diffusion models exactly?" Lacks comparison with other energy-based generative models**
>
> **A3:**
> 1. We refer the reviewer to **Appendix C in the paper, where we have already compared various generative models**, including VAE and EBM.
> 2. Simper generative models could potentially be also effective in continuous control. However, at present many research consistently indicate that **diffusion models outperform other generative methods.** ([2] (Figure 1), [3] (Table 4), and [4] (Figure 1)). They are also successfully deployed in real-world robots [1].
> 3. We conducted some preliminary experiments applying GMM to EDA, with the results below (mixture number = 100, EM training, 2 seeds):
>
> | |Average| Half-ME | Half-M | Half-MR | Hop-ME | Hop-M | Hop-MR | Walk-ME | Walk-M | Walk-MR |
> |---|--|---|--|--|--|---|--|--|---|---|
> | Diffusion BC| 67.0 | 73.9 | 47.9 | 42.2 | 71.1 | 63.9 | 69.9| 98.9 | 68.7 | 66.5 |
> | GMM BC| 63.7 | 68.2 | 45.3 | 44.7 | 56.6 | 64.3 | 67.4| 97.0 | 68.7 | 65.0 |
> | **Diffusion+EDA**| **87.3**|93.2|57.0|51.6| 104.9 | 98.4 | 92.7 | 111.1 | 87.4 | 89.2 |
> | **GMM+EDA** |28.3|1.8|40.2|36.6|3.0|33.1|2.2|47.5|62.0|28.0|
>
> [1] Octo: An Open-Source Generalist Robot Policy
>
> [2] DiffusionQL paper
>
> [3] Offline Reinforcement Learning via High-Fidelity Generative Behavior Modeling  ICLR 2023
>
> [4] Imitating Human Behaviour with Diffusion Models  Neurips 2023
>
> **Q4: The method is only evaluated on the D4RL benchmark. Additional benchmarks would strengthen the experimental validation. "**
>
> **A4:** We definitely agree that additional results from other benchmarks could enhance the validation of our experiments. However, given the limited time for rebuttal, we face some practical dilemma:
>
> 1. Our primary goal is to demonstrate the effectiveness and competitiveness of our proposed algorithm. This involves comparisons with state-of-the-art diffusion baselines in the field of offline reinforcement learning, where D4RL is the predominant benchmark. To our knowledge, there are **very few public benchmarks with a sufficient number of diffusion baselines** for meaningful comparisons besides D4RL.
> 2. Our experiments already covers **16 tasks** within the D4RL benchmarks, covering **3 distinct fields**: Locomotion, Navigation, and Manipulation. Additionally, we provide illustrative 2D experimental results in **6 more tasks**. Considering that the core contribution of this paper is somewhat theoretical, we believe this is sufficient to validate the effectiveness and support the main claims of our method.
>
> **Q5: What are the benefits of using the proposed method over alternative methods that apply DPO on diffusion models in the image generation field, such as Diffusion-DPO? "**
>
> **A5:**
> 1. Most image diffusion alignment methods, including Diffusion-DPO, are preference-based. **In offline RL, the concept of rewards is essential, and such rewards are usually continuous instead rather than binary.** Aligning pretrained policies with continuous rewards is both theoretically and practically significant.
> 2. Different from image generation, it is **more important in RL to have an efficient way for calculating data density**, which current diffusion models do not provide.  PPO/SAC/REINFORCE all require calculating data prob. The proposed BDM models introduce a novel approach to address this issue, potentially expanding the application of diffusion models in RL, as also noted by reviewer FQV2.

---

> > ### Comment · Reviewer_jAma · 2024-08-12
> >
> > I have reviewed the rebuttal, and my concern has been addressed.

---

> > > ### Author Response · Authors · 2024-08-13
> > >
> > > Thank you for your positive feedback! We are glad that our responses help.

---

### Official Review · Reviewer_2HKR · 2024-07-24

**Soundness:** 3
**Presentation:** 2
**Contribution:** 3
**Rating:** 6
**Confidence:** 2

**Summary:**

Diffusion models have shown impressive results in solving real-world problems. This paper builds on the success of large language models (LLMs) to enhance the development of diffusion models. Specifically, they introduced Efficient Diffusion Alignment (EDA), a pipeline to train diffusion models in two stages similar to LLM: pretraining and finetuning. In the pretraining stage, they proposed a bottleneck diffusion model (BDM), which modifies the score function from predicting control actions to predicting the scalar value of the control action. With this pre-trained score function, they suggest performing policy optimization by aligning the pre-trained diffusion model to Q-function values. The authors conducted experiments on various environments and compared the results to several relevant baselines. This paper's proposed EDA pipeline outperforms all the previously proposed algorithms for aligning diffusion models and provides empirical insights into why their algorithm works well. In particular, they demonstrated that their algorithm is very sample efficient, requiring very few samples to learn successfully, and they showed that their algorithm could be combined with various Q-learning methods successfully.

**Strengths:**

- The paper addresses an important problem regarding alignment with diffusion models.
- The paper performs several ablation studies to showcase the importance of key algorithm design choices.
- The policy optimization procedure to minimize the q-values between optimal and reparameterization q-function is interesting and works well in practice.

**Weaknesses:**

- The paper needs more clarity regarding the difference between behavior density estimation and the normal diffusion model.
- The paper needs more clarity about how the pre-trained scalar network is trained.
- The experiment results for other baselines are missing standard deviation bars, so it is hard to tell if the results are significant, given that some baselines are extremely close in performance.
- The value and preference optimization experiments need to include the derivation of what you optimized when k>2 for the preference experiments.
No ablation experiments compare the proposed BDM method with traditional conditional diffusion models.
- There are no vanilla BC experiments in the results presented.

**Questions:**

- Is the Behavior density estimation model a value network instead of a policy? Because you are essentially predicting a scalar value given state and action at a particular time.
- How are you minimizing equation (10) if your BDM model outputs a scalar but your Gaussian noise variable \epsislon is a vector of dimension the number of actions?
- What do you mean by the bottleneck value being expanded back to R^|A| through back-propagation? The input to the function f is an a_t, which is a scalar, not \bold{a}_t, which is a vector.
- Does f^*=\nabla_{\bold{a}_t} \log\mu_t(a_t|s,t)? If so, what is the difference between \epsilon_\phi and \f_\phi?
- Is a_{t} = \bold{a}_t * e_{i}, where e_i is a vector of the standard basis?
Should the equation on line 154 be Q^* instead of Q?
- Does f^\pi and f^\mu both output scalar values, not vector corresponding actions? If so, how are you in equation (13) outputting a vector value?
- In table 1, how does vaniila BC and vanilla diffusion BC perform?
- Line 242,  there is a typo; you meant to say figure 4, not table 4.
- On line 254, you should reference the figure in the text so the reader can correlate what the text is saying with the results in the figure.
- What is the sample efficiency of the diffusion-QL baseline? Because it is not included in Figure 5(a).

**Limitations:**

Yes

---

> ### Author Rebuttal · Authors · 2024-08-05
>
> # Official Response to Reviewer 2HKR (1/2)
> We thank the reviewer for the very detailed feedback! The reviewer's concerns are twofold: 1. confusion regarding what the BDM model is trying to model/learn and how to train it; 2. the comprehensiveness of the experiments. We hope our explanations and the newly added experimental results can help address these concerns.
>
> **Q1: Needs more clarity regarding "the difference between behavior density estimation and the normal diffusion model" and "how the pre-trained scalar network is trained. "**
>
> **A1:**
> Suppose our dataset distribution is $\mu(a|s)$.
>
> | | Classic Gaussian Policy | Normal diffusion | Bottleneck Diffuson (ours)|
> |--|--|--|--|
> | network | $\pi_\theta(s):\mathcal{S} \rightarrow \mathcal{A}$ | $\epsilon_\theta(a_t,s, t)=a_{t-1}:\mathcal{A} \times \mathcal{S} \times \mathbb{R} \rightarrow \mathcal{A}$ |  $f_\theta(a_t,s, t): \mathcal{A} \times \mathcal{S} \times \mathbb{R} \rightarrow \mathbb{R}$ |
> | predicts| $\pi_\theta^*(s)=\mu(s)$| $\epsilon_\theta^*(a_t,s, t) = \nabla_{a_t}\log\mu_t(a_t\|s)$ | $f_\theta^*(a_t,s, t) = \log\mu_t(a_t\|s) + C(s)$ |
> | predicts| average **action** | **score function** of action distribution | behavior action **density** (log probs) |
> | sample action | $a =\pi_\theta(s)$| $a_{t-1}=\epsilon_\theta(a_t,s, t) + \text{noise};a=a_0$ | $a_{t-1}=\nabla_{a_t}f_\theta(a_t,s, t) + \text{noise};a=a_0$ |
> | sampling type| direct| iterative | iterative |
> | Expressive model?| No| Yes | Yes|
> | Allow density calculation?| Yes | No | Yes|
> | training| $\|\pi_\theta(s)-a\|^2$| $\|\epsilon_\theta(a_t,s, t)-\epsilon\|^2;a_t=a_0 + \sigma_t\epsilon;\epsilon\sim\mathcal{N}(0, I)$ | $\|\nabla_{a_t}f_\theta(a_t,s, t)-\epsilon\|^2;a_t=a_0 + \sigma_t\epsilon;\epsilon\sim\mathcal{N}(0, I)$ |
>
> The **only difference** between normal diffusion models and our BDM is as follows: the diffusion model $\epsilon_\theta$ directly estimates the score function (outputting a vector of dimension $\mathcal{A}$), while BDM $f_\theta$ uses its derivative $\nabla_{a} f_\theta$ to estimate the score function. Thus, $f_\theta$ outputs a scalar, but $\nabla_{a} f_\theta$ is a vector of dimension $\mathcal{A}$. **Therefore, we can think of BDMs as exactly normal diffusion models with a different model architecture.** (See Figure 2 in the paper).
>
> The training methods remain exactly the same.
>
>
>
> **Q2: Is the Behavior density estimation model a value network instead of a policy?**
>
> **A2:** Following **A1** the BDM model $f_\theta$ is a learned behavior policy, but it can also be used to estimate the behavior density for a given action.
>
>  **Policy**: When considering $\epsilon(a_t,s,t):= \nabla_{a_t} f_\theta(a_t, s, t)$, which is easily calculatable by Pytorch through back-probagation. Then BDM is exactly a diffusion policy that can be used to sample an action. **This feature is mainly used during pretraining and evaluation.**
>
> **Value**: Without the partial derivative, $f_\theta(a_t, s, t) \approx \log \mu_t(a_t|s)$ , the BDM model is like an energy-based model that can be used to estimate action likelihood. **This is mainly used during the alignment phase.**
>
> **Q3: Does $f^\pi$ and $f^\mu$ both output scalar values, not vector corresponding actions? If so, how are you in equation (13) outputting a vector value?**
>
> **A3:** Yes, $f^\pi$ and $f^\mu$ both output scalars. We may not fully understand the reviewer's question but the output of Eq 13, including $Q_{\theta}$, $\pi_{t, \theta}(a_t|s, t)$ and $\mu_{t, \phi}(a_t|s, t)$ are indeed all scalars.
>
> Original Eq 13 in paper:
>
> $Q_{\theta}(s, a_t, t) := \beta \log \frac{\pi_{t, \theta}(a_t|s, t)}{\mu_{t, \phi}(a_t|s, t)} + \beta \log Z(s) = \beta [f_\theta^\pi(a_t|s,t) - f_\phi^\mu(a_t|s,t)] + \beta [\log Z(s, t)- C^\pi(s, t) + C^\mu(s, t)]$
>
> **Q4:Does $f^ * =\nabla_{a_t} \log\mu_t(a_t|s,t)?$ If so, what is the difference between $\epsilon_\phi$and $f_\phi$?$**
>
> **A4:** **No.** $f_\mu^* =\log\mu_t(a_t|s,t)$ while $\epsilon_\mu^* =\nabla_{a_t}\log\mu_t(a_t|s,t)$. $\epsilon_\phi$ is actually a notation for **normal** diffusion model. You can see that $\epsilon_\phi$ is rarely used when describing BDM models.
>
> **Q5: What do you mean by the bottleneck value being expanded back to R^|A| through back-propagation? The input to the function f is an a_t, which is a scalar, not a_t, which is a vector.**
>
>
> **A5:** The input $a_t \in \mathcal{A}$ of $f_\theta$  is a **vector**.  (explained in **A1** and **A2**)
>
> Stage 1: $f_\theta(a_t,s, t): \mathcal{A} \times \mathcal{S} \times \mathbb{R} \rightarrow \mathbb{R}$ takes in this vector and squeezes $a_t$ into a single scalar, which we called bottleneck energy.
>
> Stage 2:  When calculating $\nabla_{a_t} f_\theta(a_t,s, t)$, we perform back-propagation and the output is again a vector of dimension $\mathcal{A}$. **(bottleneck value being expanded back to R^|A|)**
>
> Vector -> Scalar->Vector.
>
> It is just like a U-Net. **Figure 2 (left) in the paper** is a more illustrative explanation.
>
>
> **Q6: No vanilla (diffusion) BC experiments in the results presented.**
>
> **A6:** We thank the reviewer for this suggestion and present additional experimental results below.
> ||Average|HalfCheetah-ME|HalfCheetah-M|HalfCheetah-MR|Hopper-ME|Hopper-M|Hopper-MR|Walker2d-ME|Walker2d-M|Walker2d-MR|
> |--|---|--|----|--|--|--|---|---|-------|---|
> |**EDA(ours)**|**87.3**|93.2±1.2|57.0±0.5|51.6±0.9|104.9±7.4|98.4±3.9|92.7±10.0|111.1±0.7|87.4±1.1|89.2±5.5|
> |DiffusionBC|67.0|73.9±28.5|47.9±3.8|42.2±6.5|71.1±37.1|63.9±15.0|69.9±28.2|98.9±25.1|68.7±24.5|66.5±9.9|
> |BC|43.0|35.8|36.1|38.4|111.9|29.0|11.8|6.4|6.6|11.3|
>
> *BC results come from D4RL paper.
>
> ***
>
> ## Reminder
> **QA7-QA12:**
> **We refer the reviewer to the global rebuttal posted at the top of the webpage for the rest (part 2/2) of our response. This is due to the severe page limit**.

---

> > ### Comment · Reviewer_2HKR · 2024-08-12
> >
> > Thank you for the detailed response and explanations. Since most of my concerns have been addressed, I will increase my rating to a 6.

---

> > > ### Author Response · Authors · 2024-08-12
> > >
> > > Thank you for your positive feedback! We are glad that our responses help.

---

### Author Rebuttal · Authors · 2024-08-05

# Rebuttal Summary

We would like to thank all the reviewers for their valuable comments. We are encouraged to see all reviewers recognize the theoretical novelty of our work. Reviewers FQV2 and y7D4 highlight the critical importance of the problem with diffusion models that  we are trying to solve. They also point out the vast potential of the proposed BDM model. Concerns primarily relate to the method's computational efficiency, the clarity of the paper, and limitations in the D4RL experiments.

Below, we summarize the main actions taken during the rebuttal:
1. Provided detailed computational efficiency experiment results for EDA and compared these with several diffusion-based baselines.
2. Conducted additional experiments to offer more baseline results, such as BC and diffusion BC.
3. Additionally compare with Diffusion-QL for sample efficiency.
4. Conducted an extra ablation study on various generative models (switching from diffusion models to GMMs/EBMs).
5. Rigorously revised the manuscript for a clearer narrative.
6. Clarified several confusions or misunderstandings regarding our paper.

We look forward to further discussions with the reviewers!


---

|

|

|

***


# Official Response to Reviewer 2HKR (2/2)
continue due to page limit

**Q7: Experiment results for other baselines are missing standard deviation bars**

**A7:** We did not report std bars for other baselines because:
1. It is a **common practice for most previous work** (e.g., Diffuser, QGPO, DiffusionQL, IDQL) to not report std bars of other work.  This is mainly due to the severe page limit.
2. In Table 1. The experimental results for baselines are cited from previous work. Different work may use different metrics to calculate std [1] (std, max-min, 0.5 * std, etc.). Some work (IDQL) simply do not report std.

Nonetheless, we present the results with std bar for the two highest-related work which share the same evaluation metric as ours:

|Algorithm |Half-ME|Half-M|Half-MR|Hop-ME|Hop-M|Hop-MR|Walk-ME|Walk-M|Walk-MR|Ant(U)|Ant(UD)|Ant(MP)|Ant(MD)|Kit-C|Kit-P|Kit-M|
|----|---|---|--|--|---|---|---|--|--|---|--|--|---|--|---|---|
|EDA(ours)|93.2±1.2|57.0±0.5|51.6±0.9|104.9±7.4|98.4±3.9|92.7±10.0|111.1±0.7|87.4±1.1|89.2±5.5|93.0±4.5|81.0±7.4|79.0±4.2|84.0±8.2|81.5±7.3|69.3±4.6|65.3±2.2|
|DiffusionQL|96.8±0.3|51.1±0.5|47.8±0.3|111.1±1.3|90.5±4.6|101.3±0.6|110.1±0.3|87.0±0.9|95.5±1.5|93.4±3.4|66.2±8.6|76.6±10.8|78.6±10.3|84.0±7.4|60.5±6.9|62.6±5.1|
|QGPO|93.5±0.3|54.1±0.4|47.6±1.4|108.0±2.5|98.0±2.6|96.9±2.6|110.7±0.6|86.0±0.7|84.4±4.1|96.4±1.4|74.4±9.7|83.6±4.4|83.8±3.5| | | | |

**Q8: No ablation experiments compare the proposed BDM method with traditional conditional diffusion models.**

**A8:** We note that 4 out of the 8 selected baselines in Table 1 (Diffuser, DiffusionQL, IDQL, QGPO) are based on traditional diffusion models. For instance, EDA(ours) and IDQL share the same Q pretraining method and behavior model architecture, making IDQL an appropriate ablation baseline for comparison with EDA.

Furthermore, we clarify that **traditional diffusion models cannot be used in our proposed alignment method**. These models estimate the score function but do not estimate behavior density, which is required by our loss function in Eq. 14 (see **A1**). Therefore, a direct ablation experiment is not feasible.  **This is the very motivation for us to propose the BDM model**.

**Q9: It is hard to tell if the results are significant, given that some baselines are extremely close in performance**
**A9:** We respectfully disagree with the reviewer's comment.

1. **D4RL is a highly competitive benchmark**. The selected baselines like IDQL, Diffusion-QL, and QGPO are recognized state-of-the-art methods, and EDA outperforms each by at least 2% in overall performance. Given the inherent randomness in RL tasks and the diversity across 16 tasks, it is natural for performance between EDA and other SOTA methods to be close on some tasks.
2. EDA demonstrates **significant improvements in sample efficiency and convergence speed** (Figure 5). With just 1% of the data, EDA retains 95% of performance while the best-performing baseline maintains only 80%. **This improvement should not be considered as "close performance."** Such high sample efficiency is crucial for real-world applications of alignment algorithms.
3. Furthermore, EDA clearly surpasses preference-based methods (83 vs. 76), as detailed in Figure 6.


**Q10: Sample efficiency of the diffusion-QL baseline.**

**A10:**  We conduct additional experiments studying Sample efficiency of diffusion-QL. Results are averged across 9 Locomotion tasks and 5 random seeds each.



| Algorithm      | 100%       | 10%        | 1%         |
|----------------|------------|------------|------------|
| **EDA (ours)** | 87.3 ± 3.5 | 85.6 ± 3.7 | 87.3 ± 5.1 |
| **Diffusion-QL** | 87.0 ± 2.9 | 60.2 ± 6.6 | 53.0 ± 10.7 |
| IQL            | 75.7 ± 7.7 | 65.0 ± 7.5 | 52.8 ± 7.2  |
| QGPO           | 86.4 ± 3.5 | 74.5 ± 3.7 | 71.2 ± 5.1  |

For fair comparison, we do not use the original critic training method in Diffusion-QL, but switch to the IQL-critic training method, which is consistent with ours and the IQL baseline.


**Q11: Typo in Line 242 (Figure instead of Table).**

**A11:** We appreciate the reviewer's careful reading. The typo has been corrected (though we cannot update the official paper during the rebuttal). We have reviewed the paper to ensure no similar mistakes remain.

**Q12: On line 254, you should reference the figure in the text so the reader can correlate what the text is saying with the results in the figure.**

**A12:** We thank the reviewer for the detailed suggestion. We have now ensured that all figures and tables are referenced before they are discussed in the text.

---

### Decision · Program_Chairs · 2024-09-25

**Decision:**

Accept (poster)

**Comment:**

## Summary
This paper proposes a novel two-stage approach for offline RL, called "Efficient Diffusion Alignment," inspired by alignment approaches for LLM. The paper treats the offline learning of policies as a two-stage process inspired by LLM alignment approaches:
1. Pre-training: The policy is pre-trained on large, reward-free behavior datasets over the state-action papers to estimate the behavior density.
2. Policy fine-tuning: The diffusion policy is fine-tuned with a method inspired by DPO, where they align the diffusion behavior policy with the Q-values.

The paper reports that EDA performs very well in D4RL environments and is much more sample-efficient than the baselines compared against.

## Decision
Overall, the paper is clear and easy to read. It studies an important problem: improving the sample efficiency of the offline RL algorithms. The proposed approach of using a two-stage algorithm similar to LLMs is not completely novel; a similar approach was used to train offline RL agents for Starcraft 2, which would be beneficial for this paper if the authors could cite it. However, the idea of pretraining a behavior density estimator with the diffusion policy first and then aligning them on the offline dataset with DPO is a novel and interesting approach.

The authors did a good job addressing the reviewers' criticisms of the paper. Overall, the reviewers were unanimously positive about the paper's acceptance. There were some concerns raised by the reviewers during the rebuttal and some confusion about certain parts of the paper. I would recommend the authors go over the reviews carefully and incorporate the suggestions made by the reviewers to eliminate confusion. Overall, I think this paper would be of interest to the broader AI community, and the experiments are convincing. Thus I am suggesting accept for this paper.

[1] AlphaStar Unplugged: Large-Scale Offline Reinforcement Learning, Mathieu et al, 2023: https://arxiv.org/abs/2308.03526